# A Survey on 5G Coverage Improvement Techniques: Issues and Future Challenges

**DOI:** 10.3390/s23042356

**Published:** 2023-02-20

**Authors:** Chilakala Sudhamani, Mardeni Roslee, Jun Jiat Tiang, Aziz Ur Rehman

**Affiliations:** Centre for Wireless Technology, Faculty of Engineering, Multimedia University, Cyberjaya 63100, Selangor, Malaysia

**Keywords:** cellular network, network coverage, small cells, device to device communication, non-orthogonal multiple access, optimization, multiple input and multiple output

## Abstract

Fifth generation (5G) is a recent wireless communication technology in mobile networks. The key parameters of 5G are enhanced coverage, ultra reliable low latency, high data rates, massive connectivity and better support to mobility. Enhanced coverage is one of the major issues in the 5G and beyond 5G networks, which will be affecting the overall system performance and end user experience. The increasing number of base stations may increase the coverage but it leads to interference between the cell edge users, which in turn impacts the coverage. Therefore, enhanced coverage is one of the future challenging issues in cellular networks. In this survey, coverage enhancement techniques are explored to improve the overall system performance, throughput, coverage capacity, spectral efficiency, outage probability, data rates, and latency. The main aim of this article is to highlight the recent developments and deployments made towards the enhanced network coverage and to discuss its future research challenges.

## 1. Introduction

Nowadays, the usage of wireless devices is increasing rapidly along with the amount of data consumption, user demands, and their expectations. Advances in wireless evaluation have introduced new services and use cases to meet these demands and expectations [1,2]. The essential services of these wireless technologies provide key parameters such as low latency, high bandwidth, maximum throughput, and enhanced capacity [3,4]. Fifth-generation (5G) wireless technology has been proposed to fulfill these key parameters with new optimised and enhanced services. 5G introduces new challenges such as high data rates, ultra-reliable low-latency communication (URLLC), high connectivity, greater bandwidth, and better support for mobility [5]. An evaluation of technology generations with their access techniques, data rates, frequency bands, applications, and key parameters is shown in Table 1 [6]. To meet these key parameters of 5G and beyond, new technologies are proposed, including millimetre wave (mmWave) spectrum for large bandwidths, multiple input and multiple output (MIMO) for massive connectivity, and extreme mobile broadband (eMBB) for high data rates and low latency [7,8].

The mmWave is a very underutilised spectrum, and it is used by the 5G network to provide very high-speed communications and large bandwidths. The International Telecommunication Union (ITU) identified the frequency bands for international mobile telecommunications (IMT) from the mmWave bands. The ITU identified more bands for 5G deployment at the World Radio Communication Conference 2019 (WRC-19), which took place from 28 October to 22 November 2019. They are 24.25 GHz–27.5 GHz, 37 GHz–43.5 GHz, 45.5 GHz–47 GHz, 47.2 GHz–48.2 GHz, and 66 GHz–71 GHz. As a result, the conference has identified a total of 17.25 GHz of spectrum, compared to the 1.9 GHz of bandwidth that was accessible prior to WRC-19 [9].

The frequency bands specified by 3GPP (3rd Generation Partnership Project) for 5G are frequency range 1 (FR1) and frequency range 2 (FR2), from 450 MHz to 6 GHz and 24.25 GHz to 52.6 GHz, respectively. These bands are again subdivided into different new bands. FR1’s new bands are n78 (3.3 GHz–3.8 GHz), n79 (4.4 GHz–5.0 GHz), and n77 (3.3 GHz–4.2 GHz). Out of these three bands, n78 is the most commonly used FR1 band for 5G deployment. Out of all these new bands, some are used for uplink (UL) and some are used for downlink (DL). The maximum carrier bandwidth of FR1 is 100 MHz, and the sub-carrier spacing is 15 kHz, 30 kHz, and 60 kHz. FR1 uses both time-division duplexing (TDD) and frequency-division duplexing (FDD) modes. The new frequency bands of FR2 are n256: 24.25 GHz to 27 GHz, n257: 26.6 GHz to 29 GHz, n260: 37 GHz to 40 GHz, and n261: 27 GHz to 28 GHz. FR2 uses only TDD mode for duplexing, and the maximum carrier bandwidths are 200 MHz and 400 MHz. The sub-carrier spacing is 60 kHz, 120 kHz, or up to 240 kHz.

The mmWave frequency bands are vulnerable to high attenuation due to their channel conditions and high path loss, which deteriorate the signal quality. However, the expectations of future cellular networks are high data rates, enhanced coverage and throughput, better connectivity, and massive mobility, which cannot be fulfilled without proper signal quality. The mmWave travels at high frequency and collides with the obstacles, causing a loss of signal energy. As a result of their lower penetration losses, mm-wave signals can only travel shorter distances, resulting in a much smaller coverage area than conventional frequency bands. Hence, coverage is one of the most challenging issues in the future of cellular networks.

### 1.1. Existing Surveys

According to the author, there is no detailed review of coverage enhancement techniques in cellular networks. However, there are some papers that explore the challenges of the future cellular network requirements, including coverage as one of the subtopics. In 5G, coverage is the basic issue, and to overcome this issue, many authors have proposed various algorithms and methods. To improve coverage in cellular networks, heterogeneous networks (HetNets) and small cells are proposed [10,11,12]. The HetNets with less complex user association algorithms and small cells with interference mitigation algorithms are still required to improve the efficiency and reliability of the cellular system. In the literature, network coverage was improved using optimization techniques such as machine learning, linear programming, and mathematical analysis. The optimization of coverage capacity, deployment cost, and coverage area in 5G are explored [13,14,15,16]. Along with the deployment cost and network capacity, it will provide the optimization algorithms for resource allocation, quality of service (QoS), spectral efficiency, and throughput.

The conventional multiple access techniques will not fulfil the future challenges of 5G, such as high data rates, low latency, massive connectivity, and high spectral efficiency. In 5G, the increased number of cellular users increases the receiver’s complexity. However, non-orthogonal multiple access (NOMA) techniques can be used to reduce receiver complexity [17,18,19]. This will allocate the multiple users in one resource block based on allocation techniques and decode the users at the receiver to reduce the system’s complexity. This method will improve the bit error rate, throughput, and system capacity. Device-to-Device (D2D) technology is also used to reduce the system complexity, provide security from malicious users, and provide better reliability [20,21,22,23,24]. The bandwidth of the cellular network can be improved by using carrier aggregation (CA) techniques [25,26,27,28,29]. The opportunities and future challenges of massive MIMO in 5G and beyond are discussed in [29].

In the literature, there exists several techniques and technologies to enhance the network coverage, but not in an integrated manner. The authors focused on key parameters of the 5G networks and its implementations, deployments individually. The authors in Refs. [10,11,12,13,14,15,16,17,18,19,20,21,22,23,24,25,26,27,28,29], focused on one or two coverage enhancement techniques such as small cells, CA, D2D, NOMA, MIMO, or 5G optimisation techniques—but not all in one paper. Therefore, this survey aims to provide broad and complete information about all coverage enhancement techniques and the intelligent optimization techniques, their pros and cons to fulfil the existing gap in wireless cellular networks, and also future challenging issues. A comparative overview of existing surveys on different 5G coverage enhancement techniques are shown in Table 2.

### 1.2. Contributions

This research primarily focused on cellular network coverage enhancement approaches and conducted a literature review on the topics of coverage enhancement techniques, coverage definitions, and coverage requirements. Techniques for improving coverage capacity, spectrum efficiency, throughput, latency, and data rates are analysed. The contributions of this paper are summarised as:This survey focused on the 5G network coverage. Starting from the coverage definitions, requirements, and its enhancement techniques;The evaluations and advancements of mobile networks from 1G to 5G, are discussed;3GPP releases are explored towards the coverage enhancements, spectrum bands, multiplexing techniques, number of channels aggregated, and their bandwidths;In this survey, we highlighted the 5G network key parameters, applications and advantages;The coverage degradation sources are identified and the enhancement techniques are explored;The coverage enhancement techniques such as small cells, carrier aggregation, device to device communication, non-orthogonal multiple access, multiple input and multiple output, and optimization approaches are discussed;The pros and cons and the future challenges of each enhancement technique are highlighted.

### 1.3. Paper Outline

This article is organised as follows: Section 2 illustrates the basics of wireless communication system, i.e., coverage definitions, coverage issues, coverage monitoring, and coverage requirements. Section 3 focuses on the coverage enhancement technologies such as small cells, CA, D2D, NOMA, MIMO, and optimization in 5G networks. In this section, each coverage enhancement technique implementations and their future challenges are explored. In Section 4, 5G key parameter enhancement techniques, their advantages, limitation and future challenges are illustrated. 6G and its future challenges are explored in Section 5. Section 6 concludes the article and provides the future scope. The structure of this article is shown in Figure 1. Table 3 provides the list of abbreviations used in this article.

## 2. 5G Network Coverage Planning

### 2.1. Coverage Definitions

Coverage Definition: The area covered by one base station (BS) or cell where the user can send a service request or receive a service, which is defined as coverage, is shown in Figure 2a [30]. This coverage area depends on many parameters such as transmitting power of the cell, type of antenna used, carrier frequency and the location of the cell. Coverage area is also measured in terms of the cell boundary or cell radius and is defined as the maximum distance from the cell to the user equipment (UE). The user can send any service request or receive any service without any interference within the cell boundary. In 5G, small cell site concept was proposed to avoid latency, to improve the spectral efficiency and to provide high data rates. A small cell site of 100 m cell radius is used at higher frequency bands. Therefore, to cover the large area more than one small cell site is required to be deployed. To avoid interference between two cells, i.e., out of cell interference, it is mandatory to maintain a minimum distance between the cell sites called inter site distance.

In 5G, hundreds of small cell sites are required to cover a large area and the basic cell structure is shown in the Figure 2b [30]. Based on the coverage area, the number of cells can be increased, as in tier 1, which contains 6 cells, tier 2, containing 12 cells, tier 3, containing 18 cells, and so on.

### 2.2. Identification of Coverage Issues

In cellular network, coverage is one of the most important parameters. It is affected by many parameters such as high network density, location of UE and BS, and the environmental conditions. Some of the coverage issues are coverage holes, overshoot coverage, weak coverage, channel coverage mismatch, and the cell edge problems [31]. Coverage hole is a location within the cell boundary where the UE cannot receive signals from the primary cell when it is actually in the primary cell or neighbouring cell if it is at the cell edge. This is mainly due to obstacles such as buildings, trees, mountains, etc., which will reduce the signal strength because of multipath fading effects and because the signal strength received by the UE is less than the threshold value. The coverage holes are mainly due to non-line of sight, poor radio frequency (RF) network planning, and multi-path fading.

Overshoot coverage is exactly opposite the coverage area. Overshoot coverage occurs when the signal strength is very high and it crosses the primary cell boundary. The issues with overshoot coverage are signal interference and poor quality of service. Weak coverage is the result of the signal strength being below the necessary threshold inside the cell boundary. These two issues can be mitigated by proper adjustment of RF signal power within the cell area. Channel coverage mismatch occurs due to the UL and DL frequency variations. In a cellular network the coverage issues are mainly at cell edge or cell boundary because the signal power decreases exponentially with respect to the distance [32]. To reduce the cell edge issues, sectoring and micro cell concepts were proposed [33,34].

### 2.3. Coverage Monitoring

The coverage issues mentioned in the above subsection need to be solved to provide a better cellular network. To rectify these issues it is first mandatory to identify the cause by using drive test method, performance measurements, and alarm and traces. To detect the weak coverage and overshoot coverage locations, unsuccessful handovers and terminated connections, a drive test is the best method [35]. The authors in Ref. [36], proposed a coverage mapping and cell boundary mapping techniques to solve the coverage issues. To monitor the network and to predict the network, network planning tools play an important role. ASSET radio planning tools are used to predict the network coverage and neighbouring cell planning techniques. RF planning tools [37] and RAN planning tools [38] are proposed in the literature to monitor the coverage. The coverage issues and network parameter optimization techniques are discussed in the following sections.

### 2.4. Coverage Enhancement Requirements

All cellular technologies define their requirements before their initial releases. The 5G defined its requirements such as ultra-reliable low latency communication, high spectral efficiency, high data rates, massive mobility, and coverage enhancement [39,40,41,42]. In 5G, a small cell site concept was proposed to improve the coverage but in rural areas large cell sites are still used because of low traffic rates. In suburban areas, based on the traffic density, the number of relay nodes deployed are increased to improve the coverage [43]. Therefore, based on the requirements the cell sites are deployed to improve the coverage, throughput, and spectral efficiency [44]. Mobility, handovers, and cell edge data rates also affect the coverage [45]. The authors in Ref. [46], proposed a method to reduce the data transfer rate at the cell edge and also to reduce the inter-cell interference and to improve the coverage. This section provides an overview of cellular network requirements and in the following sections the coverage enhancement techniques and their future challenges are explained.

## 3. 5G Coverage Enhancement Techniques

In this survey, small cells, carrier aggregation, device to device and NOMA, MIMO and 5G optimization techniques are proposed as the 5G network coverage enhancements techniques, which is shown in Figure 3. The existing works of each technique are discussed and its future challenges are explored in this section.

### 3.1. Small Cells

Small cells are low power short range transmission systems or BS deployed to cover a small geographical area. The characteristics of small cells are similar to the conventional BSs. These small cells can be deployed in indoor or outdoor regions and it will provide high data rates and low latency. These small cells are classified into three types based on the area covered by the BS, the maximum number of user’s covered, and the maximum power transmitted [47]. They are femto cells, pico cells, and micro cells, which are shown in Figure 4. The classification of small cells and their parameters are shown in Table 4.

5G requires small cells to achieve its key parameters such as enhanced mobile broad band, URLLC, massive connections and enhanced spectral efficiency. Small cells use low power transmitting stations and are very easy to deploy because of their low complexity in hardware installation. These small cells can be mounted on a wall for indoor applications and small towers or lamp posts can be used for outdoor applications. In small cells, the coverage and data rate depends on the operating frequency, transmitted power, and height and tilt angle of an antenna. The radius of the coverage area is always proportional to the transmitted power [48]. The small cells have many advantages such as high data rates, faster deployment, cost effective, they can work with low powers, and require less installation space. To achieve high data rates and large capacity, the 5G network is integrated with small cells, MIMO, beam-forming, and mmWaves.

The mmWave, i.e., from 30 GHz to 300 GHz, is a much less utilised spectrum and it provides very high speed communication. However, the problem with high frequency mmWave is that it collides with obstacles such as trees, buildings, etc., in the free space, and causes multipath fading, which leads to reduced signal strength. Therefore, mmWaves are the best frequency bands for short distance communication and can be used in small cells. A geographical area without small cells and with small cells is shown in Figure 5. In Figure 5a, the mmWave signal is facing multipath fading but in Figure 5b, with the use of small cells, the multipath fading is avoided and it provides better coverage than compared to the conventional method.

In the literature, researchers deployed the femto, pico, and micro cells and identified the network capacity, data rates, and coverage area [49,50,51,52,53,54,55,56,57]. In Ref. [49], the authors considered a stochastic geometry theorem for the optimization of small BS density in a ultra dense Hetnets. The optimized small BS density enhances the energy efficiency for a particular number of UEs. In Ref. [50], the authors deployed a small cell in a moving bus and estimated the coverage area as 300 m in an urban environment. For indoor applications such as residential, hospitals, schools, offices, and shopping malls, the femto cells are preferred because of their easy installation, cost effectiveness, and low power requirements [51]. The femto cells will improve the system throughput and signal to interference noise ratio (SINR) and also minimizes the latency and the cell edge coverage problems [52]. The performance of femto cells is affected by the location they are placed and the infrastructure present in the place, such as number of walls, curtains, doors, ceiling, etc. They will obstruct the signals and cause multi path propagation.. This may reduce the SINR and overall coverage area [53].

The maximum coverage radius of femto cells is up to 50 m; if the area covered by a cell increases, then the femto cells are replaced by the pico cells. The pico cells are used in indoor and outdoor regions and it also increases the coverage radius up to 200 m and the maximum power required is 250 mW [54]. If this coverage radius is extended from 200 m to 2.5 km, then the pico cells are replaced by the micro cells. A micro cell covers a large area and the power radiated by an antenna is also increased to 5 W. The pico cells are deployed at the cell edge of macro cells to enhance the coverage capacity, throughput, and traffic offload at the macro cell edge [55]. The coverage area of a macro cell is added with the pico cells to improve the overall system throughput and the received signal power is used to estimate the path loss [56]. This method will enhance the coverage capacity and cell edge spectral efficiency. A macro cell and pico cell cooperative scheduling scheme is considered to reduce the UL and DL interference due to the macro cell on pico cell UEs [57]. The problem with these small cells is handovers, i.e., frequent handovers cause major issues in the small cells. Therefore, the open research challenges in small cells are: power demands, coverage radius, deployment and testing, and mobility and handovers. Different techniques have been used in recent research to resolve the coverage and capacity issues in 5G networks with small cells, as summarised in Table 5.

The future challenges of small cells are:Power Requirements: For data transmission, the small cells use mmWave frequency bands, which require strong operating powers. A huge number of small cells must be installed in order to cover a vast area with a high population density, which requires more transmission power. Therefore, the power required will increase with the increase of small cells, which increases the deployment cost and energy. Hence, power optimization techniques must be proposed in the future to enhance the network coverage;Coverage Radius: Small cells coverage radius is less compared to the conventional BSs. The users need more coverage capacity in their working places or living places, where the small cells are to be deployed. In places where there is no need of any coverage, such as ponds, lakes, and forests, the small cells are not required to be deployed. Therefore, the deployment of small cells depends on the population density, available carriers, and topography. Along with these it also depends on small cell parameters such as supporting frequency bands, carrier frequencies, and coverage area size. The coverage radius is one of future challenges in the small cells to provide a better coverage and to avoid any gaps in the coverage;Mobility and Handovers: Small cells cover a small geographic area. When a small cell is installed in a shopping mall and a person is moving from one floor to another floor and browsing, emailing, calling, or downloading files, then it should not cause any degradation in the connectivity. Therefore, how many UEs it can handle at a time with high data rates is also one of the challenging issues to be considered in the future;Deployment and Testing: Small cells can utilise the existing infrastructure such as street lamp poles, walls, top of the apartments, etc. The weight of a small cell is one of the important factors when deploying them in indoor or outdoor regions. The correct way to test a small cell is to improve the quality of service and also to satisfy the subscribers demands. The user traffic, mobility and handovers, and overall system load should be considered as the testing parameters. These parameters should be improved by proper testing. So, AI based algorithms need to be proposed to test the small cells automatically. Identifying the correct testing method is also one of the future challenging issues.

### 3.2. Carrier Aggregation

Carrier aggregation is the process of combining one or more carriers to increase the data rate and network capacity for continuous transmission or reception [58]. Initially, the unused spectrum bands or carriers are identified and then added with the primary carriers. These combined carriers are referred to as component carriers (CC), and these are only the downlink carriers or a combination of the downlink and uplink carriers but not only the uplink carriers [59]. CC is divided into two types: primary CC (PCC) and secondary CC (SCC). The PCC is always in active mode, but the SCC can be in sleep or active mode based on the requirements of the user. If the user requires higher data rates, then the SCC is in active mode. Otherwise, it will be in sleep mode, but the PCC will always be in active mode until the handover is required. In CA, the primary serving cell or PCC, carries both the controlling signals and user data. The secondary serving cell or SCC, carries only user data; it does not carry any control information, as shown in Figure 6 [60].

In a network, if four carriers are aggregated, then one is PCC and the remaining three are SCCs. CA in 5G is divided into two types: intra band and inter band, and again these two are subdivided into contiguous and non-contiguous ways of carrier aggregation, which are classified as follows [61].

Intra-band contiguous method: One or more adjacent carriers of the same band are aggregated to form a single carrier, which is called the intra-band contiguous method of CA;Intra-band non-contiguous method: One or more carriers separated by some band gap in one frequency band are aggregated to form a single carrier. This is called the intra-band non-contiguous method of CA;Inter-band non-contiguous method: One or more carriers of different frequency bands are aggregated to form a single carrier. This is called the inter-band non-contiguous method of CA. Figure 7 [59] depicts these.

CA is introduced in 3GPP release-10 to enhance the data rates and mitigate the inter-cell interference of the LTE-advanced network [62]. In the initial stage, it can combine two or more CCs to achieve a maximum bandwidth of 100 MHz, with each CC having a bandwidth of 1.4 MHz, 3 MHz, 5 MHz, 10 MHz, 15 MHz, and 20 MHz [63]. In both intra and inter bands, the maximum number of UL or DL carriers that can be aggregated is 5. CA uses both FDD and TDD duplexing modes and also uses licensed and unlicensed bands for aggregation. The number of wireless devices increases day by day, and the bandwidth of 100 MHz is not sufficient to support the required data rates. Hence, new releases with increased bandwidths and new applications are proposed by the 3GPP. Release-13 is used by LTE Advanced Pro, and it will support a maximum bandwidth of 640 MHz and aggregate up to 32 UL and DL carriers [60]. This release allows unlicensed frequency bands in the 5 GHz range to aggregate with licensed bands. Releases 15 and 16 are proposed for 5G New Radio (5G-NR) and provide a maximum bandwidth of 6.4 GHz. Up to 16 CCs are aggregated, and those CCs can select their frequency bands from both the FR1 and FR2 bands. The CA of both LTE and 5G-NR is known as dual connectivity (DC), and it is used by the 5G headset for data transmission to achieve a maximum bandwidth of 6.4 GHz [64].

The coverage can be enhanced by using carrier aggregation techniques [65,66,67,68,69]. The high and low frequency components are aggregated based on cooperation in order to enhance the coverage and capacity in a building [65]. In Ref. [66], contiguous and non-contiguous CA methods are deployed, and the authors observe that the proposed methods improve the bandwidth and also enhance the coverage and data rates. In Ref. [67], aggregation of two CCs is considered, and the performance of CA with independent carriers is compared. The authors observed that the CA achieves a higher data rate compared with the independent carriers. In Ref. [68], single and multi-flow relay-assisted CA systems were analysed, and it was observed that the relay can be used as a capacity booster for low-frequency CCs. It will improve the capacity and fairness of the CA system. LTE and 5G-NR carriers are aggregated to enhance the coverage, channel capacity, and data speed [69]. CA has several advantages, including an unutilized spectrum that can be effectively used to improve spectral efficiency, to increase the UL and DL data rates with higher throughput, and improve the system performance. As has been summarised in Table 6, different techniques have been used in recent research to resolve the coverage and capacity issues in 5G networks with CA.

CA is facing DL, UL, and implementation challenges that need to be addressed in the future:DL Sensitivity: Instead of designing a duplexer for each carrier component, the interference between DL and UL at the receiver is used to design it. A separate multiplexer or duplexer is required if there is a large frequency separation between the UL and DL frequency bands. Therefore, designing a multiplexer for reducing RF front-end design at large frequency variations is a future challenge;Harmonic Generations: The harmonics are generated by the use of non-linear components in duplexers, power amplifiers, and transceivers. These harmonics are generated to reduce system complexity. This is one of the open challenges for researchers to design the best harmonic generator;Filter Design: In California, proper filter design is required to decode the carriers at the receiver without any interference;Power Amplifier: The intra-band UL CA signals use a higher bandwidth and a high peak-to-average power ratio (PARP) to reduce the maximum power. The maximum power can be reduced by adjusting the resource block configurations. Therefore, to achieve reduced power with high bandwidth and high PAPR, tuned power amplifiers are required. So, the open challenge is the design of tuned power amplifiers for UL CA;Implementation Issues: The hardware implementation of CA is very critical, as it requires oscillators, radio frequency chains, signal processing techniques, a strong battery life, etc. [70].

### 3.3. Device to Device Communication

The way information is exchanged between two UEs is changing as technology evolves. In mobile communication, even though two pieces of user equipment (the source and destination devices) are very close to each other, they need a central entity called BS to establish a connection between them. They are not allowed to establish a direct connection between them without BS, and it increases the data traffic at BS [71]. Therefore, to address this issue, D2D communication is proposed, and it is one of the most popular coverage improvement techniques on mobile networks [46]. In D2D communication, two devices can communicate with each other without the use of a base station. The features of D2D communication are traffic offloading, low latency, low power consumption, more spectral efficiency, and a large cellular coverage area [72].

In D2D communication, the cellular architecture is divided into two tiers. In the first tier, the communication is between the BS and the device in the macro-cell layer, called the macro-cell tier. In the second tier, also called the device tier, only devices will communicate with each other. In D2D communication, the users can share their spectrum resources in the device-tier using licensed spectrum and in the macro-cell-tier using unlicensed spectrum. The device-tier mode of communication is divided into underlay and overlay modes. The D2D users share the same spectrum resources as cellular users in in-band mode, but in macro-cell tier mode, D2D and cellular users share different spectrum resources [73]. The two-tier network is classified into four types [74].

Device relaying with an Operator Controlled Link Establishment (DR-OC): In this method, a device will communicate with the BS if there is poor cell coverage or if it is at the cell edge. This method is used to relay the data to other networks, as shown in Figure 8.

**Figure 8 sensors-23-02356-f008:**
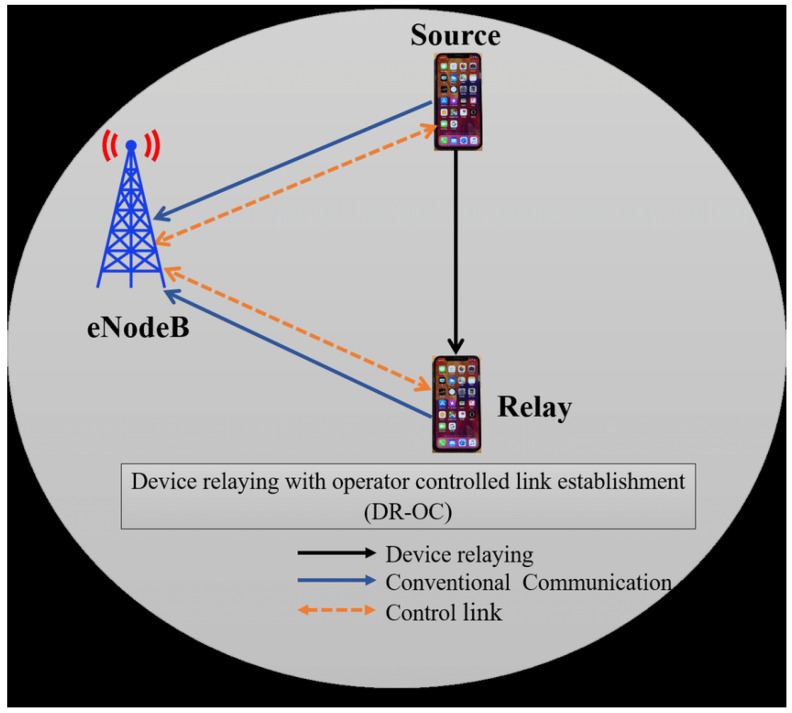
Illustration of DR-OC communication.

Direct D2D Communication with Operator Controlled Link Establishment (DC-OC): In this method, the BS will establish a link between the source device and the destination device. Once the link is established, the devices will communicate with each other directly without any role of BS, as shown in Figure 9.

**Figure 9 sensors-23-02356-f009:**
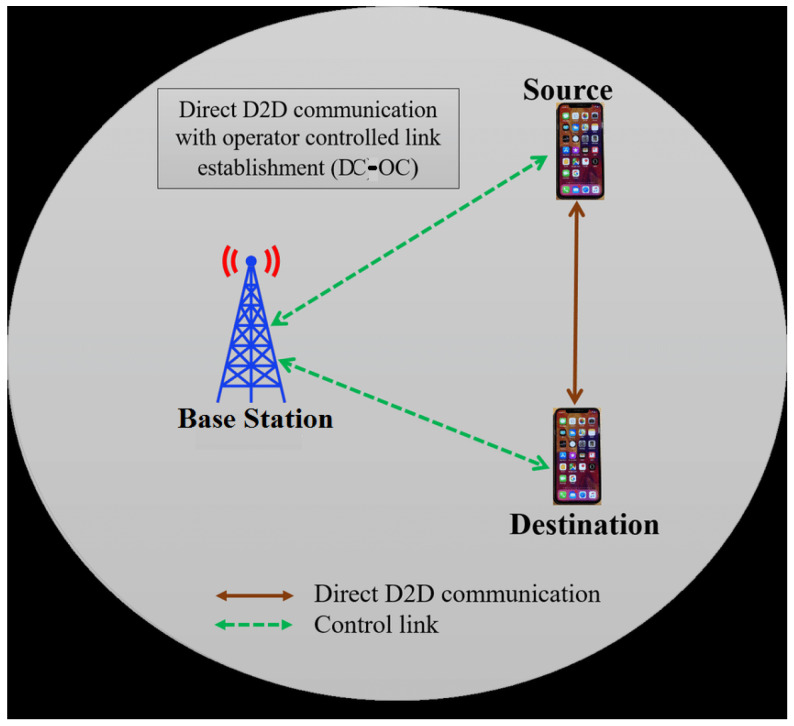
Illustration of DC-OC communication.

Device Relaying with Device Controlled Link Establishment (DR-DC): In this method, the source and destination devices will communicate with each other using relay transmission data. In this method there is no role of BS, as shown in Figure 10.

**Figure 10 sensors-23-02356-f010:**
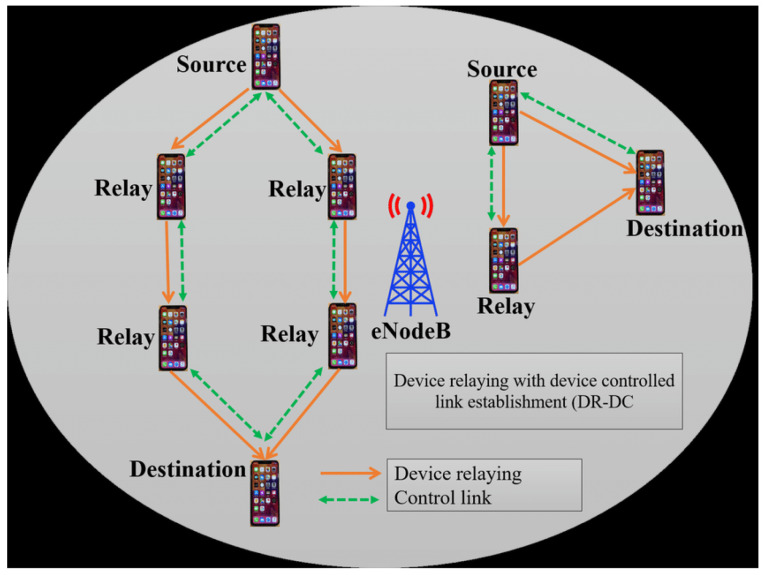
Illustration of DR-DC communication.

Direct D2D communication with Device Controlled Link Establishment (DC-DC): In this method, the source and the destination devices will communicate with each other by establishing a direct link between them on their own, and there is no role of BS, as shown in Figure 11.

**Figure 11 sensors-23-02356-f011:**
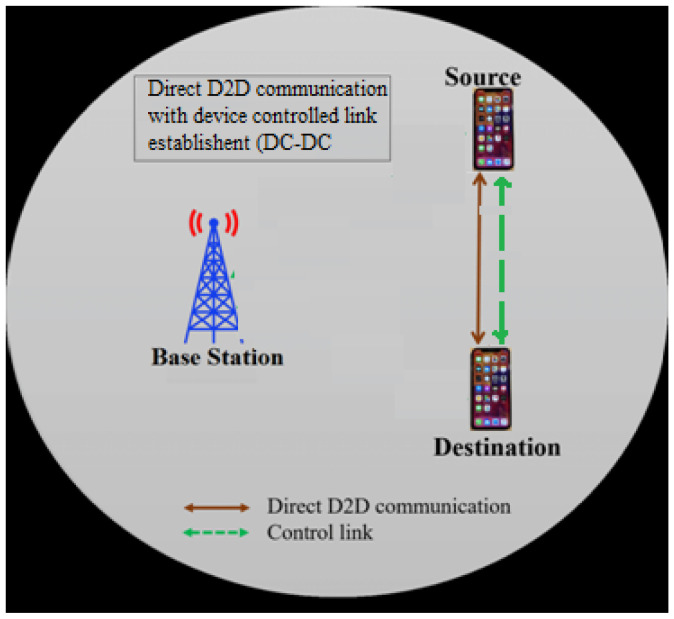
Illustration of DC-DC communication.

The successful implementation of D2D communication faces many issues and challenges such as mode selection, device discovery, interference management, and security and mobility management [75,76,77].

#### 3.3.1. Mode Selection

In D2D communication, two users can directly communicate with each other. Mode selection is the process of choosing the right mode, i.e., D2D or cellular mode, for communication between two UEs, which will improve the system’s performance, throughput, spectral efficiency, and reduce latency. The mode selection issues and proposed solutions are discussed in this section [78,79,80,81,82,83,84,85,86,87]. In Ref. [78], a joint optimization algorithm has been proposed for mode selection and power allocation in D2D underlay communication. This will improve the total sum rate and overall system performance using relay-based D2D. A degree of freedom-based mode selection algorithm and an interference alignment algorithm for interference management are proposed in Ref. [79]. These two algorithms will reduce interference and enhance the system’s performance at high SNR. These two algorithms will be suitable for large MIMO systems and small cell 5G networks. A distributed coalition formation algorithm has been proposed to achieve high system performance under link allocation and mode selection [80]. An underlying D2D multi-hop relay-aided scheme has been proposed to improve the coverage capacity [81]. A joint mode selection, relay selection, and resource allocation optimization algorithm are proposed to improve the throughput and SINR [82]. A greedy heuristic algorithm is proposed for mode selection based on QoS [83]. This method will improve the total sum rate and SINR without affecting the signalling overhead.

In Ref. [84], authors proposed an optimal mode selection and resource allocation algorithm for D2D communication in order to encourage mutual cooperation among D2D users, which also increases the throughput and efficiency. In Ref. [85], a partial CSI algorithm with a low overhead was proposed. Along with this, rate adaptation, mode selection, and user scheduling algorithms are also used to reduce intercell interference and cross-link interference. The analytical expressions for mode selection in underlay mode are derived from a single antenna at the BS. In Ref. [86], an interlay mode selection model is proposed for NOMA systems. This will use SIC decoding and power domain multiplexing to reduce interference between the users. A joint mode selection and resource allocation scheme is proposed to improve the sum rate and achieve better SIC decoding outputs. In Ref. [87], an optimal mode selection algorithm for a full-duplex CR has been proposed to improve the spectral efficiency. The crucial zone is defined to identify the operating mode of users, i.e., if the receiver is inside the guard zone, then the operating mode is considered to be a half-duplex mode. If the transmitter and receiver are outside the guard zone, then the operating mode is considered full duplex mode. Different mode selection, power allocation, and resource allocation algorithms have been used in recent research to resolve the coverage and capacity issues in 5G networks, as summarised in Table 7.

The future research challenges of mode selection in D2D communication are:Dynamic mode selection: The papers so far discussed in this section are mostly focused on the static mode selection schemes of D2D communication. Mode selection depends on the spectrum and resource availability. Therefore, an optimal mode selection scheme needs to be proposed based on the resources available;Mode selection overhead: The mode selection depends on the channel estimation, signalling, and CSI, which leads to system overhead. Therefore, the amount of overhead needs to be minimised by proposing an appropriate algorithm to improve the throughput and device lifetime in D2D communications.

#### 3.3.2. Device Discovery

In D2D communication, before any two user equipments (UE) communicate with each other, a secured device discovery process is required. In this process, the first step is to identify whether there is any device that is near another. It is possible if one device will send a discovery signal to identify the presence of other devices that are near it, and then they can form a pair by sharing the link establishment data between the devices and BS. If the two devices are discovered by each other and a secured link is established, then they can transfer their data without revealing their identities. The D2D discovery procedure is classified mainly into centralised and distributed methods [88]. A central entity known as a BS or an access point will allow all other devices to discover one another in the centralised method. After discovery, if any device wants to connect with another device, it should send the purpose of the connection to the BS. Initially, the BS will verify the channel conditions, interference control policy, and transmission power before establishing a connection. BS will also verify the SINR, path gain and the connection possibility between the two devices. Finally, it will confirm the connection between the two devices.

In the distributed method, one device can send a device discovery signal to any other device without informing the BS. In this method, the connecting devices have to verify all the channel conditions, path gain, SINR, connection possibility, and so on. This method will cause some interference, security, and synchronisation issues. In the literature, many device discovery algorithms are proposed for both centralised and distributed methods [89,90,91,92,93,94,95,96].

To detect a large number of UEs and reduce latency and collision probability in device discovery, Ref. [89] proposed a direct discovery scheme based on a random backoff procedure. In Ref. [90], a VANET-based discovery scheme has been proposed to reduce the latency in the discovery process and to enhance throughput and energy efficiency. In Ref. [91], an adaptive D2D discovery scheme has been proposed to reduce the energy consumption required for device discovery by reducing the discovery signalling messages in the network. This scheme will improve energy efficiency and throughput. In Ref. [92], a socially aware peer discovery scheme was proposed to identify malicious and trusted users. This scheme will increase the probability of identifying trusted users and reduce the number of malicious users. Therefore, the throughput increases with the increase in trusted users in D2D communication. In Ref. [93], an energy-efficient device discovery scheme has been proposed for device discovery over static and random backoff patterns. This method will increase the overall number of devices discovered for D2D communication. A mathematical model for discovering neighbourhood devices was developed in 1994 [94]. In this method, devices are discovered when they are moving, static, or out of the network.

To improve spectrum efficiency and reduce device discovery delays, a public safety full-duplex device discovery scheme was proposed [95]. In this method, there is a possibility of switching the device discovery mode from full duplex to half duplex and vice versa based on the QoS requirements. The priority in device discovery, i.e., public safety, has the highest priority as compared to random access devices, which will enhance the discovery time. In Ref. [96], a device discovery and localization scheme using UAVs is proposed for public safety systems. The MUSIC algorithm is used to estimate the accuracy, throughput, and packet error rate. Different device discovery algorithms have been used in recent research to resolve the coverage and capacity issues in 5G networks, as summarised in Table 8.

The future challenges that need to be addressed in device discovery are:Initial device discovery signal: In D2D communication, initially the device will send a device discovery signal to detect the neighbouring devices. Any malicious user can also send the discovery signal to identify trusted users and create some security issues. Therefore, to avoid security issues due to malicious users, the initial signal generation parameters such as signal data size and control data have to be secured;Multi cell device discovery: The proposed research till now has focused on single cell device discovery models. If the user is at the cell edge or dynamically changing his location from one cell region to another cell region, then multi-cell device discovery is required. Therefore, detection of multi-cell devices is one of the future challenges in D2D communication systems;Non Repudiation and Traceability: Non repudiation prevents D2D users from being denied transmission and reception of messages. Traceability is mandatory to detect the source of false messages. This is also one of the future challenges to providing a secured data transfer;Availability and Efficiency: In D2D communication systems, the availability of a device largely depends on the degree of cooperation of devices. Efficiency is the ability of a communication system to implement and operate economically. Therefore, achieving higher efficiency in device discovery is one of the future challenges in D2D communications.

#### 3.3.3. Interference Management

In a cellular network, the same spectral resources are used by both cellular users and D2D users, which will cause interference for the cellular users. The most commonly occurring types of interference in D2D communication are co-tier and cross-tier interference. In co-tier, the interference occurs between the same tier network elements, i.e., between one D2D user (DU) and the neighbouring DU, because they are located very near each other. This is due to assigning the same frequency channels and resource blocks to multiple DUEs. This co-tier interference can be overcome by proper assignment of spectrum bands and user pairing. Cross-tier interference occurs between the device network elements, i.e., DUE and CUE, which are of different tiers. This is between one CUE and one DUE or one CUE and multiple DUEs. The basic reason for cross-tier interference is sharing the same frequency channels or resource blocks with one or more D2D users.

To manage the interference between D2D users, centralised, distributed, and semi-distributed interference-controlled approaches are proposed. In a centralised method, a central entity collects the channel state information (CSI), channel quality, and interference range from all the users in the network and then manages the interference between CU and DU [97]. The main issue with this method is the significant amount of signalling overhead required for exchanging CSI and feedback, which grows exponentially with the number of DUs. This method is useful if the network size is small. In the second method, there is no need for a centralised entity to collect complete data from all the DUs. Each user will collect the data using their own methods. Therefore, the signalling overhead will be reduced and will be used for large-sized networks. The only disadvantage is the difficulty in coordinating the users. To overcome the pros and cons of centralised and distributed methods, a hybrid interference control approach is popularly used. The interference between cellular users and D2D users can be overcome by using interference management schemes [98,99,100,101,102,103,104,105,106,107].

An advanced coding and decoding scheme is proposed for interference cancellation (IC) in the CUE and DUE and to enhance the network capacity. A two-way decode and forward relay mode-based scheme is proposed in D2D communications, and M-antenna BS is used in cellular user communication to reduce interference [98]. IC and beam-forming concepts are also used to reduce the interference between the CUE and DUE in both asymmetric and symmetric cases. In Ref. [99], a multiple interference cancellation scheme (MIC) is used instead of SIC in NOMA. The MIC optimises power consumption and provides better system performance compared to the SIC receiver. In Ref. [100], the authors proposed an innovative method to overcome the interference problem by using a MIMO with zero forcing and a minimum mean square error SIC scheme to improve the performance of the NOMA system. In this method, a practical interference cancellation scheme reduces the bit error rate and improves the system performance by predicting the interference signals. In Ref. [101], the performance of cellular networks using interference cancellation and beam-forming with M-antennas at the BS is analysed. From the results, it is observed that the interference between the D2D users and cellular users is decreased and the SNR is enhanced. In Ref. [102], a guard zone-based interference mitigation algorithm is proposed based on the SIC at the BS. This method will improve the average throughput of the UE and the transmission probability.

To avoid interference between the DUEs and CUEs, interference avoidance techniques are proposed. In Ref. [103], a multi-hop D2D resource allocation scheme was proposed to avoid interference. From the simulation results, it is observed that the system’s performance in terms of throughput has increased. In Ref. [104], a massive MIMO system operating in FDD mode has been proposed to reduce the feedback overhead. In Ref. [105], the authors proposed a relay-based D2D system to improve the system’s capacity and performance by avoiding interference. The relay can use the untapped resources of the CUE to enhance the spectrum’s efficiency. In Ref. [106], a fractional frequency reuse and a blank sub-frame concept were used to avoid interference between the CUEs and DUEs. The numerical results show that the interference level of the DUEs decreased as compared to the conventional methods. This method increases the system throughput and efficiency. In Ref. [107], a dynamic distance-based algorithm was proposed to reduce the interference. Using this algorithm, outage probability, SINR, and cell density are estimated for D2D-enabled hetnets. Different interference management algorithms have been used in recent research to resolve the coverage and capacity issues in 5G networks, as summarised in Table 9.

The future challenges in the interference management that need to be addressed are

D2D in mmWave Communication: Multi-cell D2D communication is possible with the small cell concept and mmWave in 5G networks. The mmWave improves the data rates, network capacity, and latency but different interferences occur due to the penetration losses of mmWaves. Therefore, the interference mitigation algorithms for 5G mmWave need to be addressed to provide multicell D2D communication;Cell Densification and Offloading: The small cell and macro cell concepts of 5G are integrated with D2D communication to provide multi-cell D2D transmissions. Multi-cell D2D is used to improve the spectral efficiency and overall system performance of D2D communication. Therefore, the resource allocation and interference control algorithms for multicell D2D communication are challenging future issues.

#### 3.3.4. Security and Privacy

In D2D communication, device discovery was done using a centralised, distributed, and semi-distributed scheme. The centralised and distributed schemes are paired in a semi-distributed scheme to identify neighbouring users, which causes some security issues in both cellular networks and ad-hoc wireless networks. These issues will affect the availability, confidentiality, and authenticity of the devices on the network. Therefore, to overcome these issues, some privacy- and security-based algorithms are proposed [108,109,110,111].

In Ref. [108], a user sub-channel matching algorithm has been proposed to address the spoofing and interference between the CUEs and DUEs. The proposed algorithm will reduce the interference and improve the overall sum rate. In Ref. [109], an optimization algorithm has been proposed to reduce the interference due to eavesdroppers. This proposed method will reduce interference and enhance the physical layer security of CUs and throughput, but it will protect only one CU from multiple eavesdroppers. In Ref. [110], two privacy-preserving authentication protocols are proposed to provide a secured link between the devices in D2D communication. These methods will improve the efficiency and security by reducing the interference between the DUs. In Ref. [111], a secure key generation algorithm has been proposed to secure the DUs from malicious users, which leads to improved information confidentiality between the DUs. Only a few authors proposed algorithms to solve the security and privacy issues. Different security and privacy algorithms have been used in recent research to resolve the coverage and capacity issues in 5G networks, as summarised in Table 10.

The future challenges relating to privacy and security issues are as follows:Lack of Standardisation: To provide a secured connection between the DUs, there are no standard rules for data transmission, the amount of data to be transferred, feedback signalling, security policies, and so on. Therefore, the standardisation of these parameters needs to be addressed;Security Threats: A lot of security threats are happening in D2D communications. They are denial of service, man in the middle attack, impersonation attack, session hijacking, interference attack, data leakage, malware attack, free riding attack, data modification, privacy violation, forge attack, active attack on controlled data, and so on. All these need to be addressed in the future;Privacy and Security: D2D communication allows DUs to communicate directly with one another. However, identifying the trusted users and malicious users is a challenging issue. Therefore, efficient algorithms need to be proposed to identify the trusted users and malicious users before providing a secured connection;Data Confidentiality and Integrity: Data confidentiality protects the data being revealed from unauthorised users as well as preserves the user content and secures the privacy. Integrity ensures that data is not altered during the transmission. This is also one of the future challenging issues in providing an accurate data transmission;Session Key Agreement: This session key agreement is used to ensure the security of data transfer over air interface. It is mandatory to keep the session key secret to avoid any type of security issues;Non Repudiation and Traceability: Non repudiation prevents D2D users from being denied transmission and the reception of messages. Traceability is mandatory to detect the source of false messages in a D2D communication system.

### 3.4. Non Orthogonal Multiple Access

To enhance the coverage and capacity, the cellular networks use many access techniques, such as frequency division multiple access (FDMA), time division multiple access (TDMA), code division multiple access (CDMA), and orthogonal frequency division multiple access (OFDMA). The concept of orthogonality is used in these accessing techniques to reduce the interference between users, but it also reduces the number of users multiplexed to access the spectrum.

Therefore, the concept of NOMA has been proposed in 3GPP long-term evaluation release 15 as the latest multiple access technology, and it is also implemented in 5G [112]. Initially, NOMA was used with a single cell to improve spectral efficiency. The spectral efficiency is increased by increasing the number of multiplexed users accessing the single channel with different channel gains [113]. NOMA is basically divided into two types: power domain NOMA (PDNOMA) and code domain NOMA (CDNOMA). In PDNOMA, multiplexing is based on the transmitted power, and in CDNOMA, multiplexing is based on code [114,115,116]. Along with CDNOMA and PDNOMA, there are some other NOMA techniques such as signature NOMA (S-NOMA) and compressing-based NOMA (CS-NOMA) [117]. NOMA works with MIMO, cognitive radio (CR), HetNets, millimetre waves, mobile edge computing, visible light communication, vehicles for any communication, etc. This will provide high spectral efficiency, high data rates, and massive connectivity while decreasing inter-cell and intra-cell interference [118]. In NOMA, fairness in power allocation is obtained by assigning less power to strong channel gain users and more power to weak users. This power efficiency will increase the overall throughput and efficiency. NOMA can be implemented with a single cell or with multiple cells. The authors in Ref. [119] implemented the NOMA on the multi-cell and identified some issues that occurred due to the multi-cell NOMA.

By utilising the idea of high- and low-gain users, NOMA increases the throughput and spectrum efficiency by eliminating interference. It is called successive interference cancellation (SIC). A SIC receiver is one of the most important blocks in NOMA transmission, which is shown in Figure 12 [114]. The quality detection of the strongest user who has a strong signal is identified by subtracting the other user’s signal. Similarly, the weak user detects its own signal by subtracting the others’ signal, assuming noise or interference, which will improve the system’s performance by minimising the propagation effects. However, while implementing the SIC in NOMA, security issues will arise. These issues are classified as resource allocation issues, signalling issues, and security issues, which are discussed in the following subsections.

#### 3.4.1. Resource Allocation Issues

In NOMA, one resource block (RB) is shared by multiple users, and the SIC receiver is used to decode the user information at the receiver end based on the user’s channel gains. Interference between users can be avoided by choosing proper power allocation algorithms. Otherwise, resource allocation issues such as user pairing and power allocation (PA) will arise. In user pairing, the users with less power are allocated with more channel gain, and the users with more power are allocated with less channel gain to make the channel fair for all the users at the transmitter end. At the receiver end, the SIC receiver is used to decode the same. In this method, if the number of users increases, then the complexity of decoding also increases at the receiver end. This is one of the major problems with user pairing. Along with this, there is another problem, i.e., if the users with high and low gain are transformed to mid gain, then mid gain users may be paired or may not, which leads to reduced channel capacity.

To overcome the user pairing issues, optimization techniques, game theory, and machine learning algorithms are proposed in the literature. The authors proposed an optimization method while pairing two users [120,121,122,123]. To optimise the user pairing, the channel gain should not be less than the predefined threshold. A strong channel pairing algorithm (CSS-PA) was proposed to increase the system capacity and fairness in user pairing [120]. In Ref. [121], the authors used a new pairing concept, i.e., the highest channel gain users are paired with the next highest gain users, and so on. Game theory for multiple user pairing and machine learning algorithms for user pairing are proposed to reduce the channel capacity [122,123]. Different user pairing and resource allocation algorithms have been proposed in recent research to resolve the coverage and capacity issues in 5G networks, as summarised in Table 11.

#### 3.4.2. Power Allocation Issues

In the power allocation scheme, users who have more channel gain are given less power, while users who have less channel gain are given more power. In NOMA, the users with various PA are multiplexed in one RB, which will increase the system’s performance, but inefficient PA methods cause interference and also reduce the system’s performance. Therefore, proper selection of power allocation methods plays an important role and depends on many parameters such as CSI, QoS, total system power, and channel conditions.

Therefore, many optimization techniques are proposed for PA fairness [124,125,126,127,128,129,130]. The PA is optimised by maximising the maximum and average sum rates using physical layer QoS [124,125]. The optimal PA was proposed by minimising the power consumption [126,127,128]. Different PA techniques have been proposed for maximising the capacity rate and total power [129,130]. PA based on expenditure or cost is explained using graph theory [131,132,133,134,135,136,137]. In Ref. [131], the authors proposed a simulated annealing to optimize the PA and to maximise the throughput. In Ref. [132], two user concepts were considered. In this method, power is allocated to two users or only one user, whichever has a higher power gain. In Ref. [133], the authors proposed a price-based PA optimization algorithm for two users based on their QoS. The data rate is maximised using an optimization algorithm [134]. In Ref. [135], the authors proposed an auction-based game design by bidding on the transmitting power. The spectrum scarcity issues in both licensed and unlicensed bands for NOMA were discussed in Refs. [136,137].

The joint PA and user pairing problems are solved by using an optimization algorithm [138,139,140,141,142,143,144,145,146]. To identify the best multiplexed users in NOMA, a greedy algorithm, and to optimise PA, an optimization algorithm was proposed in Ref. [138]. A joint PA and user pairing optimization has been used to maximise the transmitted power and reduce the computational complexity of the NOMA system [139]. In Ref. [140], the authors proposed a proportional fair scheduling algorithm to reduce the complexity of a single carrier NOMA system. A dynamic PA scheme was used to optimise the PA in both UL and DL and also to maximise the system throughput [141]. The proposed fractional error factor method will increase the energy efficiency and also maximise the performance gain of both the UL and DL of UE [142]. A swarm optimization algorithm has been proposed to find the outage rate and average data rate of all the user pairs [143].

The spectral efficiency and error probabilities are improved by using a sort-based user pairing method [144]. This method was used to maximise the system capacity and reduce the interference. The user pairing and PA were dynamically varied to identify the system capacity of an individual and paired users [145]. The system capacity for pairing users was better than for individual users. In Ref. [146], the bit error rate is considered along with the individual and paired user’s capacities. As summarised in Table 12, various PA algorithms and joint user pairing and PA algorithms have been proposed in recent research to address the coverage and capacity issues in 5G networks.

#### 3.4.3. Signalling Issues

With the increase in wireless users in 5G, effective spectrum utilisation plays an important role. NOMA has been proposed to improve spectrum efficiency by allocating multiple users to a single RB. However, the allocation of multiple users to one RB will cause signalling issues such as CSI imperfections, latency, SIC receiver complexity, and interference management. In the literature, many authors use optimization techniques as well as analytical techniques to address all of these issues.

SIC Issues: The concept of SIC in NOMA was introduced to decode the signals of paired users at the receiver [147]. The authors used a PD-NOMA for multiplexing the users into one RB, and SIC was used to decode the users at the received PD-NOMA for multiplexing the users into one RB, and SIC was used to decode the users at the receiver. The SIC hardware implementation is very complex at high powers, but theoretical implementation is possible using Moore’s law. The performance gain of NOMA has been affected by SIC characteristics such as receiver complexity, decoding order, imperfect channel estimation, and so on [148]. The SIC receiver has been affected by the decoding of users. If the decoding order of the users does not match the multiplexed sequence, then it will cause imperfect SIC implementation. In this case, a re-transmission request is forwarded to the BS, and the whole process has been repeated again to get the perfect SIC results from the receiver. However, it takes more time to decode the data at the receiver end [149].Therefore, optimization algorithms have been proposed to avoid SIC imperfections [150,151,152,153,154]. In Ref. [150], the authors proposed an efficient PA algorithm for multi-carrier NOMA (MC-NOMA). The optimal SIC decoding value has been determined by the outage threshold of the carrier-to-noise ratio. Two algorithms, i.e., complementary geometric programming (CGP) and arithmetic geometric mean approximation (AGMA), have been proposed to optimise the total power [151]. A two-phase algorithm has been proposed to solve the non-convex resource allocation problems in NOMA [152]. Using a matching theory algorithm, the user scheduling is optimised first, and then the branch and bound technique is used to optimise PA. In Ref. [153], an efficient successive convex approximation (SCA) algorithm is proposed to optimise the sum rate. The sum rate and system utility are maximised by the SNR and the number of iterations in the MIMO-NOMA system. The iterative PA algorithm provides a maximum weighted sum rate in the NOMA system [154].The analytical techniques used to overcome signalling issues such as outage probability and CSI of NOMA in UL and DL are addressed by the authors in Refs. [155,156,157]. In Ref. [155], the authors proposed a dynamic SIC receiver concept based on the users’ received power, and the outage probability is estimated theoretically. SIC analytically estimates the statistical CSI to improve the outage probability [156]. In Ref. [157], a new decoding algorithm is used to decode the signal information at the SIC receiver, and its outage performance is verified analytically.CSI Issues: In literature, most of the authors assumed a perfect CSI at the receiver. However, in NOMA, the concepts of SIC decoding, user pairing, and PA depend on imperfect CSI. This leads to computational complexity, overhead signalling, and delayed feedback. To overcome these issues, optimization and analytical methods are identified, and with these methods, the outage probabilities are increased by assuming imperfect CSI and also reducing the number of feedback bits [158,159,160,161,162]. The minimum outage probability is obtained from a minimum feedback rate. An optimal power allocation scheme is used to improve the total sum rate of a NOMA system [158]. The outage probability is minimised by optimising the dynamic PA using the one-bit CSI feedback method [159]. Imperfect CSI and perfect CSI models are used to estimate the system throughput and bandwidth [160]. The outage probability based on channel feedback is estimated analytically [161,162]. As summarised in Table 13, various optimization and analytical algorithms have been proposed in recent research to resolve signalling issues and improve coverage and capacity in 5G networks.

#### 3.4.4. Security Issues

In NOMA, multiple user messages are superimposed on a single RB. The SIC receiver is used to decode the overlaid signals on the receiver. In the SIC receiver, the strong users are allocated with lower powers and the weak users are allocated with stronger powers for channel fairness. At the receiver, strong users have to subtract the low-power signals to get their own signals. However, it causes some security issues, which have to be considered in NOMA. These security issues can be overcome by using encryption and decryption schemes at the sender and receiver, respectively. This process increases the latency and processing requirements. However, the basic idea of NOMA in 5G is to decrease the latency and improve the data rates, spectral efficiency, and bandwidth.

The NOMA implementation causes some security issues, such as the implementation of SIC, transmitted power, and outage probability. These security issues mainly occur at the physical layer. To overcome these security issues, optimal and analytical solutions are proposed in the literature.

Optimization Methods: The authors in Refs [163,164,165,166,167] proposed optimization techniques to overcome the security issues in NOMA technique. An optimization algorithm is used to maximise the security sum rate (SSR) in a secured physical layer of a NOMA system. The authors in Ref. [163] proposed an optimal PA and optimal power splitting algorithms to maximise the security sum rate. The proposed algorithms significantly improve the system performance by assuming a uniform PA and a fixed power splitting scheme. In Ref. [164], the authors proposed two mobility models, such as a random way point (RWP) and random direction (RD), to observe the security performance of the physical layer. The average security sum rate of NOMA users was estimated and analysed. From the numerical results, it is observed that the RWP achieves a high safety rate than all other mobility models. In Ref. [165], the security sum rate is maximised by using a two-stage optimization technique. In the first stage, the SINR of a particular user is fixed to its maximum value, and then it is used to maximise the security sum rate by using a one-dimensional search algorithm. The optimal beamforming technique is used to improve the safety rate by adjusting SINR between signal strength and interference. In Ref. [166], the authors considered the security outage probability (SOP) to maximise the NOMA security performance. Design parameters such as PA, decoding order, and data rate are estimated even though the CSI is not perfectly known at the transmitter. Pairing a strong user with an unstructured weak user improves the physical layer performance [167]. The optimization algorithm minimises the pairing outage probability. The mathematical expressions for outage probability and secured outage probability are derived based on the proposed optimization algorithm;Analytical Methods: The physical layer security is estimated using analytical methods along with optimization methods. The numerical expressions for SOP are derived using analytical methods in Refs [168,169,170]. NOMA users and eavesdroppers are randomly placed, and a protected area is generated around the source of the eavesdroppers with an imperfect SIC. The security performance of the physical layer is improved by increasing the eavesdroppers’ execution area. This is extended for multiple antenna transmissions and estimates the numerical expressions for diversity order, which will reduce the SOP of multiple antenna transmissions [168]. In Ref. [169], the authors assumed a new concept where BS and users are operating in half-duplex mode and eavesdroppers are in full-duplex mode. To interrupt the NOMA transmissions, the eavesdroppers are performing active jamming and passive eavesdropping. A novel transmission outage probability scheme is proposed to improve the outage transmission probability. The analytical expressions for SOP and security diversity orders are derived to improve the security performance. The security performance of an overlay CR-NOMA system was described in Ref. [170]. In this method, we consider the primary user (PU) as a secured user and the secondary user (SU) as an eavesdropper. This proposed system provides guarantees of QoS for the PUs and reduces the interference between the users. The analytical expressions for connection outage probability, SOP, and security throughput of the PU’s are derived by assuming the Nakagami-m fading channel. By reducing the number of SUs, security performance can be improved. As summarised in Table 14, various optimization and analytical algorithms have been proposed in recent research to resolve security issues and improve coverage and capacity in 5G networks.

#### 3.4.5. NOMA—Future Challenges and Research Issues

NOMA can be included with the other 5G enabling technologies such as MIMO, CR, HetNets, millimetre wave, and so on. This helps in achieving the 5G key parameters such as improved spectral efficiency, low latency, high data rates, extended coverage capacity, and mass connectivity. However, the inclusion of NOMA with 5G and beyond 5G leads to some issues and challenges such as resource allocation, pairing, decoding, security, and signalling. These issues can be overcome by using some technologies and techniques such as optimization and analytical techniques, game theory, and machine learning algorithms. There are still some challenges and issues with the implementation of NOMA that need to be addressed. The major concerns that need to be addressed in the future are:User Pairing in NOMA: In the NOMA implementation, multiple users are multiplexed into one RB, and the SIC receiver will decode the paired users at the receiver. We have discussed pairings of two or three users in previous sections, but not multiple users. Nowadays, the demand for connected devices such as IoT, V2V, and massive machine types is increasing, which requires them to pair a large number of users to one RB. Therefore, NOMA requires some new pairing techniques to fulfil the requirements of future wireless networks. Along with this, there is another challenge to be addressed, which is the practical implementation of user pairing in NOMA systems;Receiver Complexity in SIC implementation: In NOMA, the users are multiplexed at the transmitter side using pairing techniques and decoded at the receiver using a SIC receiver. As the number of users increases, the allocation of transmission power to pair the users becomes very complex, and at the same time, decoding the strong user to the next strong user and up to a weak user is a very difficult task at the receiver end. This process increases the latency and also introduces interference with the increased number of multiplexed users. Therefore, reducing the latency and interference and providing an efficient and dynamic SIC receiver for the NOMA system is a future challenge;Multi-cell NOMA system: In NOMA systems, most of the researchers considered a single cell, and very few addressed the multi-cell concept because the multi-cell system causes inter-cell interference. The interference will affect the performance of the weak users who are at the cell edge. Therefore, to solve the interference issues along with the pairing and decoding issues, a small cell concept of 5G is included in the multi-cell NOMA system. This is the future challenge in the multi-cell NOMA systems to be addressed;Mobility in NOMA: NOMA is used to enable 5G and beyond 5G technologies. In IoT, V2V, and M2M communications, mobility is the key parameter. Most of the research in NOMA systems is based on static systems. The PA, pairing, and SIC receiver algorithms are proposed based on the static behaviour of the users. However, in future communications, dynamic PA, pairing, and SIC algorithms will be required. As the user moves from one location to another, the channel gains vary with respect to the user’s location. Therefore, proposing dynamic algorithms for NOMA systems is one of the challenges of the future;CSI in NOMA: Most of the research is carried out assuming perfect CSI in NOMA systems, but fewer users consider imperfect CSI. The CSI plays an important role in user pairing and decoding for users with SIC receivers in the NOMA system. To increase the system performance and spectrum efficiency of a dynamic system, the estimation of CSI is very important. For a dynamic channel, the estimation of CSI using machine learning and game theory algorithms is one of the important future challenges in NOMA systems.

### 3.5. Multiple Input and Multiple Output

The key parameters of 5G can be enhanced with the use of MIMO technology. Exploiting the bandwidth and spectral efficiency features of wireless networks will help increase the throughput. The present research is concentrated on methods that increase spectral efficiency because increasing the bandwidth has the disadvantage of decreasing the SNR for the same transmitted power. Utilizing numerous antennas at the transceivers is a well-known method of enhancing spectral efficiency [171]. To provide high-speed transmission with a minimal quality of service, MIMO communication systems have been proposed based on the usage of an antenna array at the transmitter and receiver [172].

Foschini [173] and Telatar [174] observed a sharp, linear rise in the channel capacity with the increasing number of antennas in the MIMO systems. MIMO systems use two unique dimensions, i.e., diversity and capacity of the radio link, to enhance the system’s performance. Diversity enhances the communication channel’s dependability by utilising multiple antenna links, and capacity can be increased by employing multiple antennas and multiplexing techniques, which maximise the amount of information transmitted through the channel. However, the increase in data traffic these days increases the need for spectrum for 5G wireless networks and beyond. Therefore, to enhance the data rates and the channel capacity, a mm-wave spectrum with a huge number of small radiating elements at the BS, called massive MIMO (M-MIMO) technology, has been proposed [175].

Massive MIMO is an advancement in MIMO technology that increases the spectral efficiency, energy efficiency, data rates and throughput by using hundreds or even thousands of antennas connected to a BS, which is shown in Figure 13 [29]. M-MIMO is a significant technique used to enhance the throughput, spectrum efficiency, data rates, capacity and overall system performance [176,177,178,179,180,181]. In Ref. [176], the authors introduced a M-MIMO technology for 5G and beyond. The system performance, throughput and data rates are enhanced through the proposed technology and they achieved a maximum data rate of 100 Gbps. M-MIMO, in conjunction with the planar antenna array, is used to enhance the system capacity. However, the planar antenna array reduces the coverage radius to only tens of kilometres. So, to enhance the coverage radius up to 100 km, a cylindrical M-MIMO system has been proposed [177]. The proposed system enhanced the coverage radius along with the system capacity by 2.1 times compared to the conventional system with a planar array.

The beam multiple access technique was used to enhance the system capacity of a M-MIMO system [178]. This will reduce the bit error rate and enhance the system capacity by 10 times and energy efficiency by 100 times compared to conventional methods. In Ref. [179], the authors proposed a cell-free MIMO UL receiver with zero forcing, MMSE, and maximum ratio combining detectors to enhance the spectral efficiency. When compared to conventional methods, the zero forcing and MMSE methods improve spectral efficiency. In Ref. [180], the authors considered a triangular lattice structure of antennas within the frequency bands of 24.25 GHz and 29.5 GHz for M-MIMO systems. This structure achieves higher spectral efficiency by maintaining a minimum distance between the antennas. In Ref. [181], the authors estimated the optimal antennas by using the trade-off between energy efficiency and spectral efficiency in a M-MIMO system. This will reduce the cost of the transmitted power and the total energy consumption. In Refs. [182,183], the authors discussed MIMO, its opportunities, recent trends, and advantages in 5G networks and beyond 5G.

Different antenna arrays, multiplexing techniques for M-MIMO have been proposed in recent research to enhance the data rates and capacity in 5G networks, as has been summarised in Table 15.

#### MIMO—Future Challenges and Research Issues

5G key parameters such as improved spectral efficiency, energy efficiency, high data rates, and enhanced capacity are achieved using the MIMO technology. There are still some challenges and issues in the implementation of M-MIMO that need to be addressed. The key issues that need to be explored in the future are:Signal Detection: The UL signal detection becomes more complex in M-MIMO systems due to the large number of radiating elements, which decreases the system throughput. Along with this, the superimposition of user-transmitted signals at the BS causes interference, which also reduces the spectral efficiency and throughput. Therefore, to enhance the spectral efficiency and throughput of a M-MIMO UL, intelligent algorithms are required. Designing less complicated and accurate algorithms for UL signal detection is one of the future research challenges;Channel Estimation: For a perfect CSI, the M-MIMO system performance increases linearly with the number of transmitting and receiving antennas. The BS should know how to use CSI in order to identify and detect the user-transmitted signal at the UL and to precode the signals at the DL. Channel estimation at the DL and UL depends on the FDD and TDD duplexing modes.FDD Mode Channel Estimation: In this mode, the CSI needs to be estimated for both DL and UL. During the DL, the BS forwards the pilot signals to the user, and the user replies with the estimated CSI to the BS. Similarly, during the UL, the BS estimates the CSI using the orthogonal pilot signals forwarded by the user. The DL channel estimation for a M-MIMO system with a large number of antennas becomes very difficult and is impossible to carry out in real-world applications.TDD Mode Channel Estimation: The problem during the DL channel estimation in FDD mode is solved by using TDD mode. In this mode, with the advantage of channel reciprocity, the BS estimates the DL channel with the use of UL CSI. During DL, the user will forward the orthogonal pilot signals to the BS, and the BS will estimate the CSI to the user terminal based on these pilot signals;Pilot Contamination: In M-MIMO systems, the BS requires the user terminal’s channel response in order to estimate the channel. When the user terminal delivers orthogonal pilot signals to the BS, the BS estimates the uplink channel. Additionally, the BS calculates the downlink channel to the user terminal using M-MIMO’s channel reciprocity property. The BS can accurately estimate the channel if the pilot signals in the home cell and nearby cells are orthogonal. However, because there are only a limited number of orthogonal pilot signals available in a given bandwidth and period, adjacent cells must reuse the orthogonal pilot signals, which leads to pilot contamination. Hence, the pilot signal contamination is one of the challenges for future research in M-MIMO systems;Energy Efficiency: Energy efficiency is defined as the ratio of spectral efficiency to the total transmitted power. M-MIMO systems offer significant energy efficiency by achieving higher spectral efficiency with low power consumption. The increased number of antennas in the M-MIMO system will increase the spectral efficiency, but it will also increase the total power consumption, which reduces the energy efficiency. Numerous studies have been done to develop energy-efficient M-MIMO systems based on the trade-off between energy efficiency and spectral efficiency and also to reduce the power consumption by designing the power amplifiers. Therefore, achieving higher energy efficiency is one of the challenges of future research.

### 3.6. 5G Optimization Algorithms

Nowadays, cellular networks are becoming increasingly challenged to provide high data rates, low latency, enhanced coverage, and capacity. The user requirements are to be fulfilled with advanced technologies. These technologies might affect network complexity and also cost. Therefore, optimization algorithms are required to reduce the network complexity and to enhance the key parameters of 5G and beyond 5G. The benefits of optimization algorithms include reduction in interference and latency, enhanced SINR throughout the cell, spectral efficiency, throughput, coverage, and capacity [184]. In the literature, researches have proposed optimization techniques to enhance the key parameters of 5G [185,186,187]. In Ref. [185], a genetic optimization algorithm is used to identify the location of a mobile station in a cell and the location of the new BSs in the test regions. This method will enhance the coverage capacity and data rates and decrease latency and traffic density. In Ref. [186], a game theory based optimization algorithm is proposed to enhance energy efficiency and throughput based on link fairness. In Ref. [187], the authors proposed an innovative scheduling algorithm to reduce the delays and call drop rates in order to enhance the QoS.

The position of an antenna, its tilt angle and height will affect the coverage of a cellular system. Therefore, optimization of antenna parameters will be considered to enhance the coverage of a network [188,189,190]. In a cellular system, directional antennas with higher order sectorization are considered to enhance the coverage. The antenna tilt will improve the coverage and also SINR [188]. In Ref. [189], a reinforcement learning (RL) algorithm is used to improve the spectrum efficiency and the sum rate of a network, using a simplified path loss model. In Ref. [190], a self-tuning algorithm is proposed to adjust the tilt of an antenna. Two fuzzy logic controllers are used for automatic adjustment of the antenna tilt, which will improve the SINR at the cell edge and spectrum efficiency. In Ref. [191], artificial neural network (ANN) and stochastic learning based optimization algorithms are used to optimise the antenna tilt. Network coverage and capacity are enhanced by using the ANN algorithm. In Ref. [192], two machine learning algorithms are proposed to reduce the interference and to enhance the coverage by optimizing the transmitted power and antenna tilt.

The random distribution of the traffic across a cell and its uncertainty in the radio propagation needs dynamic antenna tilts and optimization algorithms to enhance the coverage and capacity. Different optimization algorithms have been proposed in recent research to enhance the coverage and capacity in 5G networks, as has been summarised in Table 16.

#### 5G Optimization—Future Challenges and Research Issues

5G key parameters such as improved spectral efficiency, low latency, high data rates, and extended coverage capacity are achieved using the optimization algorithms. There are still some challenges and issues in the implementation of AI-based optimization algorithms that need to be addressed. The key issues that are to be explored in the future are:ML and RL Algorithms: New ML and RL based optimization algorithms are required to enhance the 5G network coverage and capacity and also to bridge the gap between intelligent algorithms and 5G technologies;Scheduling Algorithms: Scheduling algorithms need to be proposed for 5G networks to optimize the throughput and to enhance the capacity of cellular networks;Optimization of the Antenna Parameter: Optimization of antenna tilt angle, antenna height, and number of antennas is required to enhance the coverage and capacity and to reduce cell edge interference, deployment cost, and complexity.

5G key parameters such as coverage capacity, latency, throughput, spectral efficiency, data rate, outage probability, interference management, and power consumption can be enhanced by using different algorithms and methods in the literature. Table 17, lists the various key performance indicators and the articles in which they were used for performance evaluation. Statistical analysis of 5G key parameters is shown in Figure 14.

## 4. 5G Enhancement Techniques—Key Technologies, Advantages, Limitations, and Future Challenges

In this section, 5G key parameter enhancement techniques, their advantages, limitations, and future challenges are explored. In Section 3, all 5G enhancements techniques were discussed. Their advantages, limitations, and future challenges have been summarised in Table 18.

5G key parameters can be enhanced using small cells, CA, D2D, NOMA, MIMO, and optimization techniques. Each technique’s pros and cons and their future challenges are listed in Table 17. In small cells, the future challenges can be achieved by developing path loss models using ML, AI, and deep learning techniques. These path loss models will enhance the coverage radii, mobility, handovers, etc. Future challenges in CA can be met by developing dynamic frequency carrier aggregation techniques and heuristic optimization algorithms to maximise battery life. In D2D, the future challenges are achieved by proposing ML and AI-based algorithms for device discovery, synchronisation, mode selection, etc. In NOMA, significant optimization techniques are required to enhance the power efficiency, user pairing, security, and receiver complexity. In MIMO, ANN and CNN algorithms need to be proposed to detect the signals and estimate the channel information. The 5G optimization algorithms are required to enhance the coverage capacity, data rates, and latency.

## 5. 6G and Its Challenges

Beyond 5G and 6G network applications, users need more network capacity, throughput, data rates, spectral efficiency, low latency, and an extended coverage area compared to 5G networks. It will transform the nation into automation, i.e., smart production, a smart society, and smart lives, by 2030 [193].

In the literature, several studies have focused on 6G networks [194,195,196,197,198,199,200,201]. In Ref. [194], a thorough analysis of 6G is provided based on a review of 5G advances, covering technology trends and issues with the goal of addressing the problems with coverage, capacity, user data rates, and mobility of mobile communication systems. In Ref. [195], authors provided a detailed overview of the 6G network and highlighted the use cases and applications of the 6G network in various dimensions. The key parameters and future research challenges of 6G are discussed. In Ref. [196], the authors focused on the use cases and technologies of 6G networks. Sixth-generation enabling technologies and their applicable use cases are compared. In Ref. [197], the authors provided the major issues and challenges relating to the security, privacy, and trust problems in 6G networks. In Ref. [198], the authors discussed the security and privacy challenges that may emerge with the 6G requirements, novel network architecture, applications and enabling technologies including physical layer security, distributed AI/ML, Visible Light Communication, THz bands, and quantum communication. In Ref. [199], the authors discussed 6G security and privacy issues in the four key aspects such as distributed AI, real time edge computing, 3D intercoms, and intelligent radio.

It is necessary for 6G capabilities to be realised at scale for extremely immersive applications such as 3D communication, virtual reality, and digital twins [200]. The advantages of the upcoming 6G technology for processing real-time applications on edge networks have been discussed [201]. The challenging issues and future directions of 6G based edge computing have also been discussed. The future challenges of 6G networks that need to be addressed are quantum machine learning for 6G communications, ultra-reliable communications for edge computing, energy harvesting for extended battery life, and so on.

## 6. Conclusions and Future Scope

This survey explored the evaluation of mobile networks from 1G to 5G, its requirements, and the key parameters of 5G, including coverage definitions and coverage requirements. It addressed the coverage enhancement techniques in 5G such as small cells, CA, D2D, NOMA, MIMO, and optimization in 5G. Each technique’s advantages, limitations, and future challenges are understandably illustrated. Afterwards, we explored the new requirements of coverage enhancement techniques such as interference management, spectrum management, device discovery, mode selection, CSI requirements, security and privacy, network deployments, and testing issues. Further, we have provided the pros and cons of existing techniques and their future challenges for enhanced coverage in 5G networks. Our paper shows the accurate path for coverage enhancement solutions and its future requirements.

One of the main goals of this paper was to provide readers with the most recent and relevant information on coverage enhancement techniques for 5G networks and beyond. To this end, we focused on the most recent studies in the field and aimed to demonstrate the current research focus within the community. We believe that the information and insights presented in this paper will be valuable to researchers working in the field of 5G wireless communications and beyond.

The use of AI in cellular networks towards SDN, effective deployment of edge computing, implementation of RAN, network slicing management, network visualization, THz communication, visible light communication, quantum communication, virtual reality, and security and privacy issues are the future research challenges of 5G networks and beyond 5G.

## Figures and Tables

**Figure 1 sensors-23-02356-f001:**
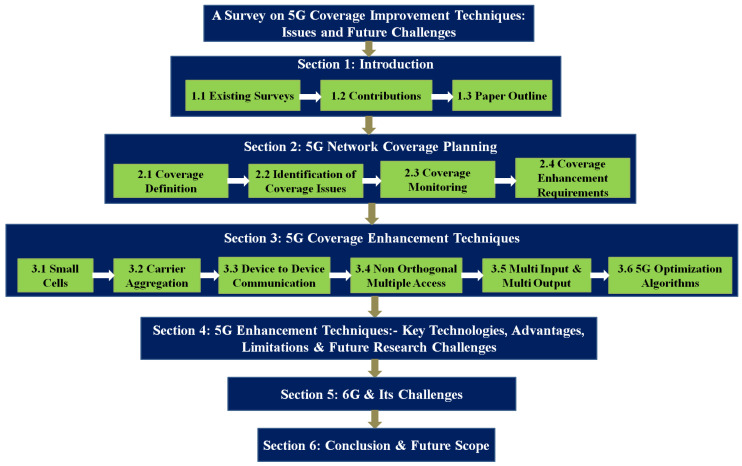
Structure of the article.

**Figure 2 sensors-23-02356-f002:**
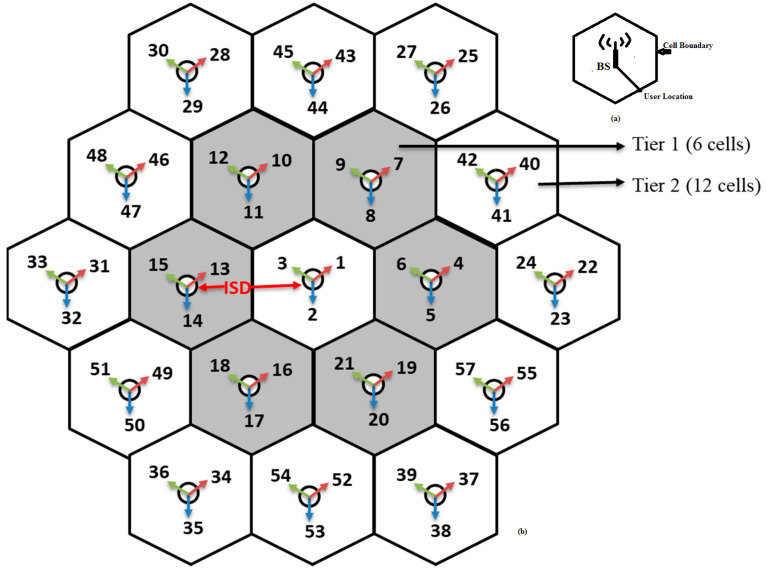
(**a**) Single cell, (**b**) cell structure.

**Figure 3 sensors-23-02356-f003:**
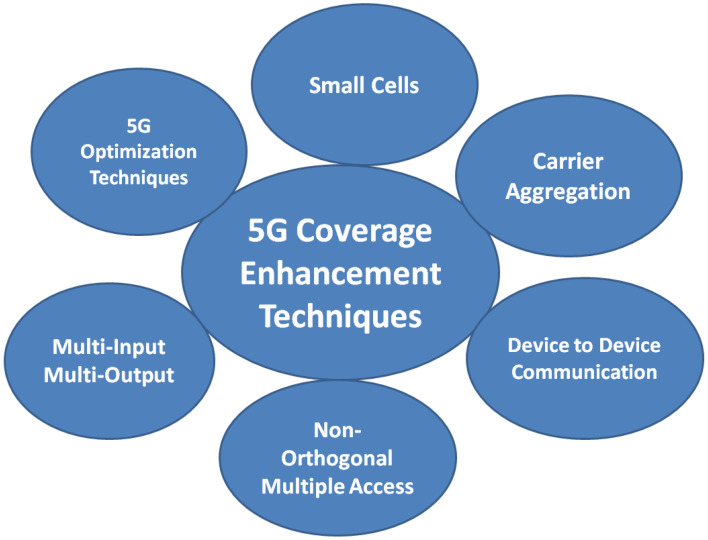
5G coverage enhancement techniques.

**Figure 4 sensors-23-02356-f004:**
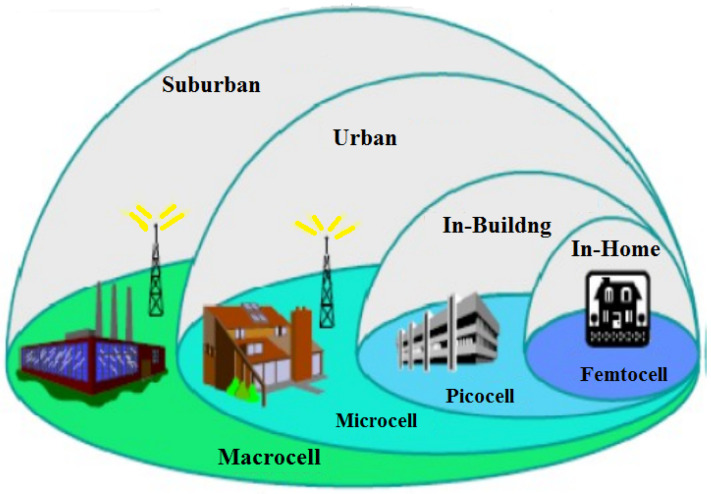
Small cell techniques.

**Figure 5 sensors-23-02356-f005:**
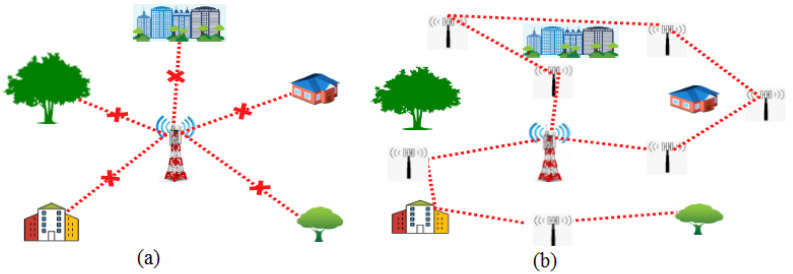
(**a**) The mmWave signal is blocked by obstacles. (**b**) Small cells used to avoid the multipath fading.

**Figure 6 sensors-23-02356-f006:**
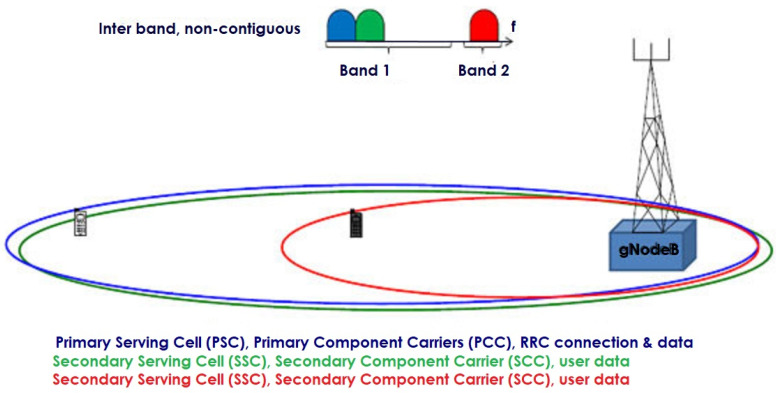
Primary carrier components and secondary carrier components in a network.

**Figure 7 sensors-23-02356-f007:**
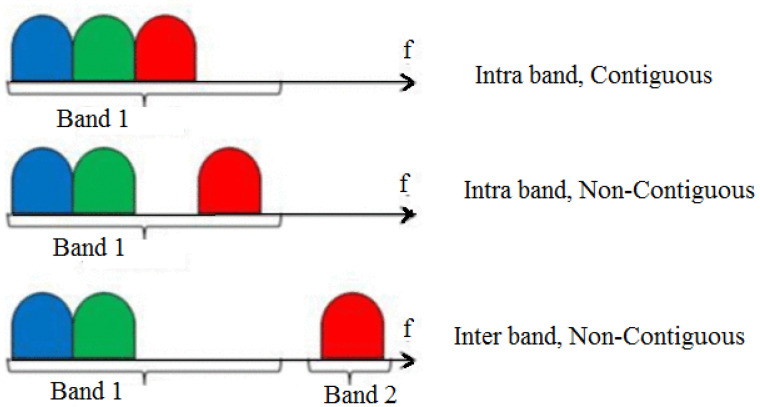
Types of carrier aggregations.

**Figure 12 sensors-23-02356-f012:**
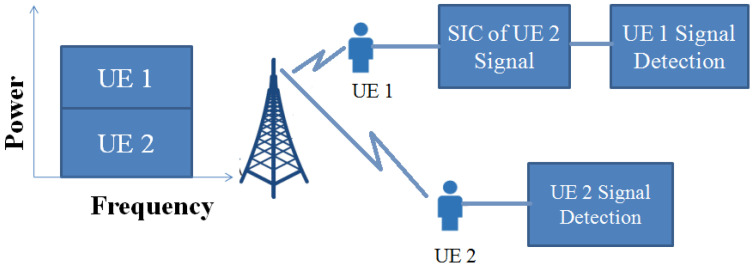
NOMA with a SIC receiver.

**Figure 13 sensors-23-02356-f013:**
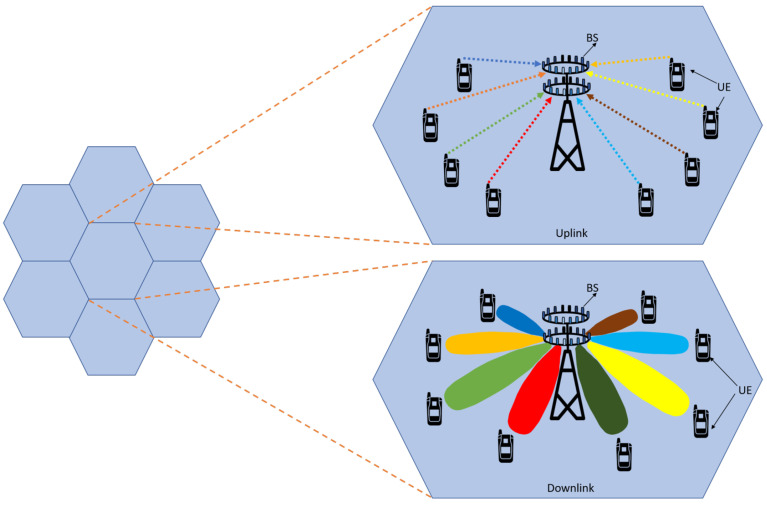
Massive MIMO with UL and DL.

**Figure 14 sensors-23-02356-f014:**
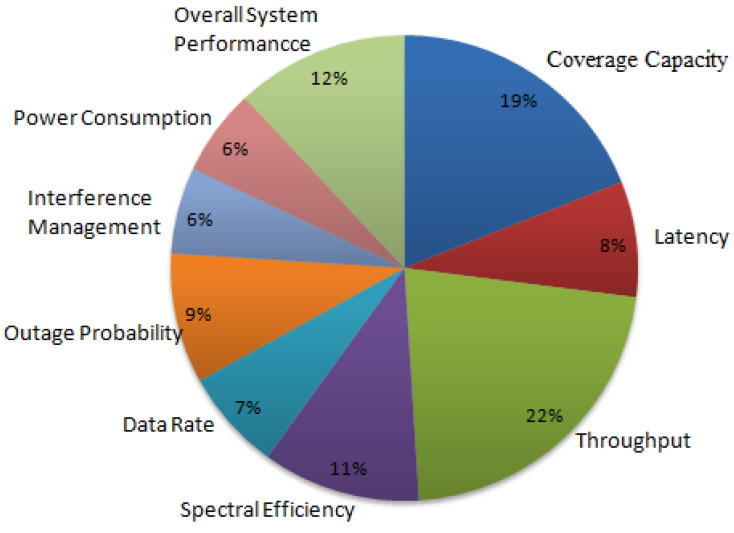
Statistical analysis of 5G key parameters.

**Table 1 sensors-23-02356-t001:** Evaluation of technology generations from 1G to 5G.

Generation	Access Techniques	Data Rate	Frequency Bands	Applications	Key Parameters
5G	NOMA, FBMC	2.4 Kbps	1.8 GHz, 2.4 GHz, 30–300 GHz	Voice, Data, Video calling, ultra HD video, virtual reality applications	Ultra-low latency, ultra-high availability, ultra-speed, and ultra-reliability
4G	LTEA, OFDMA, SCFDMA, WIMAX	10 Kbps	2.3 GHz, 2.5 GHz, 3.5 GHz	Voice, data, video calling, HD television, and online gaming	Faster broadband internet and lower latency
3G	WCDMA, UMTS, CDMA	384 Kbps to 5 Mbps	800 MHz, 850 MHz, 900 MHz, 1800 MHz, 1900 MHz, 2100 MHz	Voice, data, and video calling	Broadband internet and smart phones
2G	GSM, TDMA, CDMA	100 Mbps to 200 Mbps	800 MHz, 900 MHz, 1800 MHz, 1900 MHz	Voice and data	Digital
1G	FDMA, AMPS	10 Gbps to 50 Gbps	800 MHz	Voice	Mobility

**Table 2 sensors-23-02356-t002:** A comparative overview of existing surveys on 5G coverage enhancement techniques.

Authors & Ref. No.	Small Cells	CA	D2D	NOMA	5G Optimization	MIMO
Kim, T.Y. et al. [10]	✓	X	X	X	X	X
Ramazanali, H. et al. [11]	✓	X	X	X	X	X
Xu, Y. et al. [12]	✓	X	X	✓	X	✓
Khatib, E.J. et al. [13]	X	X	X	X	✓	X
Dash, S. et al. [14]	X	X	X	X	✓	X
Lieira, D.D. et al. [15]	X	X	X	X	✓	X
Shayea, I. et al. [16]	X	X	X	X	✓	X
Abuin, A. et al. [17]	X	X	X	✓	X	✓
Akbar, A. et al. [18]	✓	X	X	✓	X	✓
Hussain, MD. et al. [19]	X	X	X	✓	X	X
Ansari, R.I. et al. [20]	X	X	✓	X	X	X
Celik, A. et al. [21]	X	X	✓	X	X	X
Sedidi, R. et al. [22]	X	X	✓	X	X	X
Wang, M. et al. [23]	X	X	✓	X	X	X
Gandotra, P. et al. [24]	X	X	✓	X	X	X
Afolalu, O. et al. [25]	X	✓	X	✓	X	X
Alam, M.J. et al. [26]	X	✓	X	X	X	X
Chikhale, D. et al. [27]	X	✓	X	X	X	X
Lin, P. et al. [28]	X	✓	X	X	X	X
Qamar, F. et al. [29]	X	X	X	X	X	✓
Our Survey	✓	✓	✓	✓	✓	✓

Note: In Table 2, ✓ denotes covered topics and X denotes uncovered topics.

**Table 3 sensors-23-02356-t003:** List of Abbreviations.

Abbreviation	Full Form	Abbreviation	Full Form
3GPP	Third Generation Partnership Project	LTE-A	Long Term Evaluation Advanced
5G	Fifth Generation	ML	Machine Learning
AGMA	Arithmetic Geometric Mean Algorithm	MIC	Multiple Interference Cancellation
ANN	Artificial Neural Networks	MIMO	Multiple input Multiple Output
BS	Base Station	mmWave	Milli-Meter Wave
CA	Carrier Aggregation	NOMA	Non Orthogonal Multiple Access
CC	Carrier Component	OC	Operator Control
CDNOMA	Code Division NOMA	OFDMA	Orthogonal Frequency Division Multiple Access
CDMA	Code Division Multiple Access	OMA	Orthogonal Multiple Access
CGP	Complementary Geometric Programming	PA	Power Allocation
CR	Cognitive Radio	PDNOMA	Power Domain NOMA
CS-NOMA	Compressing NOMA	PCC	Primary Carrier Component
CSI	Channel State Information	QoS	Quality of Service
CU	Cellular User	RAN	Random Access Network
CUE	Cellular User Equipment	RB	Resource Block
D2D	Device to Device	RD	Random Distribution
DL	Down Link	RF	Radio Frequency
DR	Device Relaying	RL	Reinforcement Learning
DC	Direct D2D Communication	SINR	Signal to Interference Noise Ratio
DU	D2D User	SIC	Successive Interference Cancellation
DUE	D2D User Equipment	SOP	Security Outage Probability
eMBB	Enhanced Mobile Broad Band	SCC	Secondary Carrier Component
FDD	Frequency Division Duplexing	SU	Secondary User
FDMA	Frequency Division Multiple Access	TDD	Time Division Duplexing
FR1	Frequency Range 1	UE	User Equipment
FR2	Frequency Range 2	UL	Up Link
IC	Interference Cancellation	URLLC	Ultra Reliable Low Latency Communication
IMT	International Mobile Telecommunication	UAV	Unmanned Aerial Survey
IoT	Internet of Things	V2V	Vehicle to Vehicle
ITU	International Telecommunication Union	V2X	Vehicle to Anything
LTE	Long Term Evaluation	WRC	Worlds Radio-communication Conference

**Table 4 sensors-23-02356-t004:** Comparison of small cells.

Parameter	Femto Cell	Pico Cell	Micro Cell
Maximum coverage radius	50 m	250 m	2.5 Km
Maximum transmitted power	100 mW	250 mW	5 W
Maximum number of users	16	64	200
Type of backhaul	Fibre, Wired	Fibre, wired	Fibre, wired, microwave
Cost	Low	Low	Medium
Advantages	Offload network congestion, extended coverage and high data rates	Extended throughput and coverage	Extended coverage
Applications	Residential	Small enterprise	Smart cities, smart metro
Location	Indoor and outdoor	Indoor	Outdoor

**Table 5 sensors-23-02356-t005:** Summary of small cell-based approaches of 5G networks.

Authors & Ref. No.	Approach	Advantages	Limitations
Sun, Y. et al. [49]	The energy efficiency of an ultra dense HetNets is estimated based on the stochastic geometry theorem.	The small cells are optimised and enhanced the energy efficiency.	The energy efficiency is constant with a further increase of UEs and it reduces the overall system performance. UE density optimization is required to enhance the efficiency.
Feteiha, M.F. et al. [50]	Small cells are deployed in a public transport bus to verify the traffic to and from the macro cells. Traffic offloading is verified dynamically and the DL outage probability expressions are derived.	The overall system coverage and outage probability gain are enhanced. Power consumption is reduced.	The pre-coding signals are to be perfectly formatted to avoid any security issues.
Khan, M.F. et al. [52]	The impact of outdoor UEs on the traffic offload is analysed based on the cell densification concept.	Enhanced the UEs and cell edge throughput, overall system capacity, and coverage.	The overall deployment cost increases due to the deployment of additional pico cells.
Namgeol, O. et al. [54]	The interference effect of the HCS networks and the outage probability of femto cells are analysed.	Enhanced the outage probability and spectral efficiency of a HCS network.	Carrier to interference ratio between the femto cells in the HCS network to be avoided.
Landstrom, S. et al. [55]	Pico cells are deployed with the macro cells to improve the overall system performance.	Enhanced the coverage capacity and cell edge coverage. Traffic density at the macro cell is reduced.	Identifying the fixed locations of pico cells is very important in order to reduce the traffic density at the macro cell and to improve the cell edge coverage.
Yasir, B.A. et al. [56]	The coverage area of a macro cell is added with the pico cells to improve the overall system throughput and the received signal power is used to estimate the path loss.	Enhanced the cell edge spectral efficiency, coverage capacity, and throughput.	The system complexity and deployment cost increases with the number of pico cells deployed.
Lopez-Perez, D. [57]	A macro cell and pico cell cooperative scheduling scheme is considered to reduce the UL and DL interference due to the macro cell on pico cell UEs.	Enhanced the traffic offloading gain, DL throughput to 36%, UL throughput to 7% and overall system performance by reducing the interference.	Need to improve the UL throughput and reduce the interference between the pico cell UEs and macro cell.

**Table 6 sensors-23-02356-t006:** Summary of carrier aggregation-based approaches of 5G networks.

Author & Ref. No.	Approach	Advantages	Limitations
Wei, M. et al. [65]	Theoretically analysed the high and low frequency CC aggregations: 2.5 GHz and 3.5 GHz frequency bands are aggregated to get enhanced coverage capacity.	The UL and DL coverage capacity was enhanced inside and deep inside the buildings.	This method can be extended to mmWave frequency bands.
Parikh, J. et al. [66]	Contiguous and non-contiguous CA methods are used in LTE-A systems that are compatible with the LTE systems to achieve the high data rates.	Achieves high data rates, better coverage, and wider bandwidths	LTE devices which will not support the LTE-A will use only 20 MHz carriers instead of 100 MHz carriers.
Alotaibi, M. et al. [67]	The impact of spectrum aggregation (SA) on LTE networks is measured. The performance of intra band SA and contiguous CA in LTE are compared.	The spectrum aggregation of LTE system provides better system performance compared to individual carrier systems.	Lower frequency bands are aggregated for a limited cell size. For an extended cell size, high frequency components need to be aggregate.
James, A. et al. [68]	Intra and inter band CA techniques are analysed and the energy efficiency of single and multi-flow based relay assisted CA systems are proposed to estimate the ergodic rate of users.	The relays boost the coverage capacity of the lower CCs and provide a better capacity fairness between the users.	There is further need to improve the capacity fairness between the users.
Cao, Y. et al. [69]	Two LC based lumped π networks and TL based phase shifters are used with the BAW duplexers.	Throughput and overall system performance are improved.	Interference free frequency multiplexing schemes are to be used to improve the CA performance

**Table 7 sensors-23-02356-t007:** Summary of mode selection, power allocation, and resource allocation algorithms in D2D communications.

Author & Ref. No.	Approach	Advantages	Limitations
Hoang, T.D. et al. [78]	A joint optimal mode selection and resource allocation and power allocation schemes are proposed for a D2D undelayed cellular networks.	The system sum rate is improved through the optimization of mode selection schemes. An optimised resource allocation scheme reduces the co-channel interference.	D2D links will suffer from the co-channel interference from the cellular links.
Chou, H.J. et al. [79]	Degree of freedom based mode selection and interference alignment based interference management schemes are proposed.	Provides high system sum rate and low interference between DUE and CUE. The sum rate increases with the increased number of MIMO antennas.	The sum rate is low for a lesser number of antennas, i.e., it achieves only 10 bits/s/Hz for six antennas.
Yang, L. et al. [80]	An optimization model for resource allocation based on non-transferable utility and distributed coalition formation algorithms are proposed.	Traffic offload, system performance, and average sum utilities are enhanced.	A single cell scenario is used in this paper but a multi cell scenario is required in 5G systems for better coverage.
Gui, J. et al. [81]	DL throughput optimization techniques are proposed for D2D communications. The multi hop relay aided in-band D2D, game based power adjustment and greedy algorithms are discussed.	DL throughput, energy efficiency, and coverage capacity are enhanced using the proposed algorithms.	The co-channel interference increases with the increased number of UEs.
Liu, M. et al. [82]	Optimization of power allocation, resource allocation, and mode selection based relay assisted D2D communication system is considered.	Power allocation complexity is reduced and overall system throughput is maximised.	At low powers the interference between the DUs is increased and the system sum rate is reduced.
Bithas, P.S. et al. [83]	Proposed a Markov chain theory for single user and greedy heuristic algorithm for multi user scenarios for mode selection in D2D communication.	Outage probability, system sum rate, and average user SINR are improved without increasing the system complexity and signalling overhead.	At low SINR, the DUE is unable to select the best mode of operation and the system performance is affected by the synchronisation errors.
Hou, G. et al. [84]	Resource allocation and mode selection algorithms are proposed to enhance the system performance.	Spectral efficiency and system throughput are enhanced.	Multiplexing resources increases the interference between the UEs.
Bulusu, S. et al. [85]	Proposed a throughput optimal joint mode selection user scheduling algorithm and discrete rate adaptation scheme for D2D communication system.	User fairness increases with the optimal mode selection. Cell throughput increases with the multi user diversity.	Joint rate power adaptation modes for partial CSI are to be addressed.
Dai, Y. et al. [86]	D2D enabled NOMA cellular network resource allocation and mode selection issues are discussed. The expression for optimal resource allocation was derived based on game theory approach.	Enhanced the system sum rate, coverage capacity, and spectral efficiency.	Inter cell interference increases with the increased number of BSs.
Haider, N. et al. [87]	To protect the PUs, an optimal mode selection algorithm is analysed.	The success probability increases and interference between the DUs is reduced.	Optimal guard zone radius is required to improve the system capacity.

**Table 8 sensors-23-02356-t008:** Summary of device discovery algorithms in D2D communications.

Author & Ref. No.	Approach	Advantages	Limitations
Zhang, J. et al. [89]	A random back-off procedure is used to detect a device in D2D communication and analytical expressions for average delay in device discovery are derived.	Enhanced the device discovery probability. The maximum number of devices discovered are 61.4% and the success probability is 13.6%. The occurrence of collisions are reduced in the guard zone of a device discovery.	There is a trade-off between the success probability of CUs and DUs.
Chour, H. et al. [90]	A VANET based device discovery scheme has been proposed to offload the D2D discovery traffic. Analytical expressions for latency in device discovery is derived.	VANET enhanced the D2D discovery process efficiently. Achieved low latency in the device discovery process and reduced the traffic due to routing messages of device discovery.	The proposed model is only applicable to some particular geographic regions such as small roads and urban areas, but not to highways.
Mosbah, A.B. et al. [91]	An adaptive device discovery scheme has been proposed to derive the analytical expression for user density prediction. This method predicts the present device discovery performance based on the historical data.	Improved the device discovery performance using ML and optimization algorithms.	The user density changes dynamically and the prediction based method will not provide an accurate performance of the system at each and every time.
Long, Y. et al. [92]	A peer-device discovery scheme based on the deep neural networks has been proposed to detect the malicious devices. Trusted and malicious devices can be identified easily using this method.	This method will improve the efficiency of detecting the trusted devices.	Channel capacity and SNR are influenced by the instability of the device connections.
Kallem, Z. et al. [93]	Device discovery maximisation algorithm based on the conversion of half duplex mode of operation to full duplex mode using the SINR threshold was proposed.	The device discovery rate is significantly increased with the full duplex mode compared to the conventional half duplex mode.	The power consumption increases with the increased number of devices discovered. The collision probability also increases with the increased number of resource blocks assigned per user.
Jaffiry, S. et al. [94]	Proposed a neighbourhood device discovery scheme for detecting the moving UEs. This method will estimate the number of devices discovered, which are moving in a straight line between two points.	Device discovery rate increases with the device density, if the devices are moving with a constant speed in a given path.	If the devices are moving with different speeds or are randomly distributed or randomly change their locations then the proposed method fails to detect the devices.
Kallem, Z. et al. [95]	Proposed a public safety priority based in-band full duplex frame structure. A time efficient resource allocation scheme is used to convert a half duplex mode of operation to a full duplex mode when the half duplex mode fails to support the available resources.	The device discovery rate is enhanced by 37% compared to the random access method. The number of devices discovered also increased by 13.3% by reducing the collision rate.	This method provides priority to the public safety, but fails to detect the neighbouring devices, which will reduce the overall device discovery rate.
Masood, A. et al. [96]	Proposed a device discovery and localisation with the use of UAVs in the public safety networks. The MUSIC algorithm is used to locate the UEs in disaster situations.	The devices affected by a disaster are effectively detected. Enhanced the throughput and reduced the packet error rate.	The packet error rate increases with the increased SNR and with the different modulation and coding schemes.

**Table 9 sensors-23-02356-t009:** Summary of interference management algorithms in D2D communications.

Author & Ref. No.	Approach	Advantages	Limitations
Gandotra, P. et al. [98]	Proposed a NOMA based multi interference cancellation scheme using the optimal resource allocation and signal detection. Optimised the power consumption.	The proposed scheme reduces the overall system complexity up to 13 dB compared to the OFDM and conventional NOMA schemes.	Latency is increased and the fairness factor is decreased with the increased number of D2D pairs.
Wu, Y. et al. [99]	Proposed a Lunius system to optimise the local file sharing between the peer to peer users. It improves the user experience and offloads the traffic at BS in a D2D communication system.	Throughput, data rates, and spectral efficiency are maximised and the traffic load at the BS is reduced.	The Lunius system performance depends on the cooperation between the D2D links and the error rates.
Ni, Y. et al. [100]	MIMO with minimum mean square error SIC and zero forcing scheme was proposed to enhance the performance of a NOMA system. A practical IC scheme is used to reduce the bit error rate.	Data rates are increased and the bit error rates are reduced.	Only a two-users scenario is considered but in practical cases the number of users in a given cellular region are more than two.
Vu, H.V. et al. [101]	A joint beam-forming and PA schemes are used with the MIMO cellular networks to enhance the overall system throughput.	Enhanced the spectral efficiency gain and throughput in a full duplex mode of operation compared with the half duplex.	Increased D2D links compressed the spectral efficiency gain.
LV, S. et al. [102]	A guard zone based D2D scheme is proposed with the SIC at the BS of a cellular network. An analytical expression for the successful transmission probability for a uniform distribution of CUEs and DUEs is derived.	The success probability and throughput are maximised by optimising the inner radius of the guard zone.	The success probability and average throughput decreases with the increased inner radius of the guard zone.
Melki, L. et al. [103]	Multi hop D2D communication system is considered to improve the system performance.	Throughput, SINR and coverage capacity are maximised with the number of users.	This method is applicable for short range D2D links only.
Behjati, M. et al. [104]	A new IC technique is designed to reduce the feedback overhead in a massive MIMO system.	CSI feedback overhead is reduced. Overall system performance and throughput are enhanced.	As the number of antennas increases in a MIMO system, the throughput decreases.
Xue, J. et al. [105]	Proposed a relay based D2D communication system with the interference cancellation scheme.	The overall system performance and user capacity are enhanced with the relay trigger and relay selection algorithms.	The increased number of relays required to cover a large number of CUs increases the system complexity.
Kim, J. et al. [106]	A fractional frequency reuse and almost blank sub-frame schemes are proposed to mitigate the inter cell interference between the CUEs and DUEs.	The throughput is enhanced up to 60% and the interference is reduced using the frequency reuse method.	The co-channel interference between the DUE pairs is not considered in this paper.
Kamruzzaman, Md. et al. [107]	A dynamic D2D algorithm is proposed to reduce the interference based on the outage probability, SINR and cell density.	User capacity is maximised for a small cell density. Outage probability increases with the increased SINR.	The overall success probability decreases with the small cell density.

**Table 10 sensors-23-02356-t010:** Summary of security and privacy algorithms in D2D communications.

Author & Ref. No.	Approach	Advantages	Limitations
Wang, L. et al. [108]	In a D2D underlaid communication system, physical layer security is discussed and an optimal threshold is used to enhance the security of a CU.	The proposed scheme effectively protects the CUs from the eavesdroppers.	In this method only one CU is protected from the multiple eavesdroppers and D2D pairs. However, in practical multiple CUs, protection is required.
Wang, M. et al. [109]	Proposed a key agreement and privacy preserving authentication protocols to provide the secured D2D communication.	Provides a secured D2D communication from the malicious users and also protects from the internal attacks.	Estimation of success probability is further required.
Waqas, M. et al. [110]	Relay based cooperative trusted and semi-trusted and non trusted environments are used to provide a physical layer security. The optimal relay pairs are selected based on the coalition game theory.	The trusted environment will provide a better security when compared with the semi and non trusted environments.	The average user security decreases with the distance threshold.
Gupta, R.K. et al. [111]	A secured key is generated through a bit string based on the enhanced identity encryption scheme. The leakage of UE’s identity can be reduced with this method.	The time required for encryption and decryption process is much less. The UE’s identity can be hidden by using the Lagrange coefficient.	As the number of UE’s messages increases, the time required for encryption, decryption, and authentication also increases.

**Table 11 sensors-23-02356-t011:** Summary of user pairing and resource allocation algorithms in NOMA systems.

Author & Ref. No.	Approach	Advantages	Limitations
Zhang, H. et al. [120]	Two user pairing and access algorithms are proposed to enhance the system capacity of a SIC based NOMA system.	The proposed methods enhanced the system throughput, capacity, and spectral efficiency.	At minimum SNR, the average channel capacity is very small for fixed PA algorithms compared to all the other algorithms.
Ding, Z. et al. [121]	Fixed PA and CR inspired NOMA schemes are proposed to improve the performance of NOMA systems.	Enhanced the outage probability and data rates of the NOMA system with the increased SNR.	If two neighbouring users are paired then it will provide a better data rate compared to the non-neighbouring users.
Wang, S. et al. [122]	Multi carrier NOMA with cooperative game and conventional neural network schemes are used to design a user pairing network.	Enhanced the average channel throughput based on the soft and hard channel capacities with SNR and with the number of users.	As the number users increases the performance loss also increases, which degrades the average spectral efficiency.
Guo, F. et al. [123]	Interference-aware user grouping scheme is considered in the NOMA system to minimise the energy consumption. Exchange league concept is used to identify the energy consumption in the case of multi user scenario.	Computational complexity reduced with the proposed algorithm. Total transmit power varies with the number of user groups and the number of individual users. Transmit power increases with the individual users but decreases with the user groups.	Interference between the users in one group will increase randomly with enhanced number of users in that group. So, the total transmit power reduces with the user groups.

**Table 12 sensors-23-02356-t012:** Summary of power allocation algorithms and joint user selection and PA algorithms in NOMA systems.

Author & Ref. No.	Approach	Advantages	Limitations
Li, C. et al. [131]	Simulated annealing technique is considered to optimise the power allocation and to enhance the system throughput.	The proposed simulated annealing provides better throughput compared to numerical optimization methods. It also reduces the time complexity for achieving enhanced throughput.	An equal power allocation scheme is used as an optimal power allocation in this paper.
Wang, Z. et al. [132]	A novel price based PA algorithm is considered for a DL-NOMA system with two special cases such as two users scenario and multiple users scenario.	The proposed algorithm maximised the revenue of a BS up to 14.94% and the total sum rate of the users up to 20.63%.	Power fairness is required to encourage the users with low channel conditions.
Wang, Z.Q. et al. [133]	Stackelberg game theory was used to solve the power allocation problem in NOMA system with QoS constraints.	Optimal price based scheme improved the BS revenue and the number of users allowed to use power.	Power fairness is required to encourage the users with low channel conditions.
Aldebes, R. et al. [134]	Glicksberg game theory based PA algorithm is considered for the DL-NOMA system to derive an optimal expression for maximising the utility function.	The average data rate and sum rate are enhanced with the proposed scheme.	As the number of users in a cell increases beyond the threshold, the average data rates are reduced.
Lamba, A.K. et al. [135]	An auction game based PA scheme was proposed for the hybrid NOMA-OMA system. In this method, BS will acts as an auctioneer and sell the powers to the user pairs formed with one strong user and one weak user.	This method achieves very high sum rates compared to the conventional NOMA system.	User pairing plays an important role in this method to provide a power fairness.
Datta, S.N. et al. [138]	An optimal PA and an efficient user selection algorithm was proposed to maximise the performance gain of a DL-NOMA system. Karush–Kuhn–Tucker greedy algorithm was used for optimal PA and user selection.	The performance of the proposed optimal power allocation scheme is better than conventional NOMA schemes and also provides the maximum weighted sum rate and performance gain.	The inter/intra cell interference is estimated for an ideal channel and an ideal IC scheme is used to mitigate the interference.
Mei, J. et al. [139]	A joint user pairing and PA optimization scheme is proposed to maximise the system performance and to reduce the system complexity.	System complexity is reduced by 62% and the average user throughput is enhanced by 82%.	User pairing complexity is reduced but the multi user scheduling complexity is still high.
Liu, F. et al. [140]	A low complexity algorithm is designed for a joint PA and user selection scheme.	Overall system complexity is reduced, and cell edge and overall system throughput is enhanced.	In this method only two to three users are paired. If the number of users paired increases then it increases the system complexity.
Ali. M.S. et al. [141]	A joint optimization problem for maximisation of sum throughput is considered based on the power budget, SIC receiver constraints, and minimum rate requirements.	The throughput of UL/DL NOMA system is maximised.	If the cluster size increases then the performance of DL-NOMA system is reduced.
Chen, L. et al. [143]	A joint user selection and PA scheme is considered for the UL-NOMA system to improve the user fairness.	Enhanced the fairness performance, average data rates, and average outage probability.	The fairness index decreases with the user density.
Shahab, M.B. et al. [146]	A dynamic user pairing and PA scheme is considered in a NOMA system to improve the data rates and BER.	System capacity is maximised by satisfying the individual user data rates and BER.	BER is reduced up to some optimal PA value. However, it increases after the optimal PA value.

**Table 13 sensors-23-02356-t013:** Summary of optimization and analytical algorithms used to resolve the signalling issues in NOMA systems.

Author & Ref. No.	Approach	Advantages	Limitations
Wei, Z. et al. [150]	A sub-optimal iterative resource allocation (RA) algorithm is considered to provide a perfect balance between the overall system sum rate and its complexity.	Enhanced the total power saved with the increased number of users and reduced the total power consumption. The spectral efficiency gain is enhanced by reducing the overall power consumption.	The outage probability is low at low transmit powers.
Li, S. et al. [151]	Proposed a robust optimization scheme for DL-NOMA system. A non-convex problem is converted into a convex problem by using a complementary geometry programming and arithmetic geometric mean approximation schemes.	The proposed scheme achieves high transmit power savings compared to the non robust scheme. Overall power efficiency is enhanced by the robust scheme.	Average transmit power is improved if the number of users are facing poor channel gain issues and do not satisfy the minimum QoS requirement.
Cui, F. et al. [152]	An outage probability based user scheduling scheme was proposed for a DL-NOMA system under imperfect SIC. The non-convex RA problem was solved by optimising the user scheduling through a matching theory algorithm and branch and bound technique.	The proposed method minimised the total transmit power required.	The required transmit power increases with the residual interference. The increased residual interference deteriorates the outage probability of the users.
Alotaibi, S. et al. [153]	Two imperfect CSI schemes such as channel distribution information and channel estimation uncertainty are considered for the MIMO-NOMA system to enhance the sum rate and also an efficient SCA algorithm is used to optimise the sum rate.	The overall system sum rate and system utility are maximised.	Due to the imperfect CSI, the overall interference increases and there is a trade-off between system complexity and throughput.
Wang, X. et al. [154]	Imperfect SIC is considered for a DL MC-NOMA system to maximise the weighted sum rate. A low complexity PA algorithm is proposed based on the perfect and imperfect SIC.	The weighted sum rate decreases with the SIC errors and power factor.	As the number of users or number of iterations increases, the weighted sum rate also increases, which increases the total power required.
Gao, Y. et al. [155]	A dynamic decoding order SIC receiver is considered for the UL-NOMA system. Analytical expression for outage probability is derived based on the three user scenario.	The outage probability of user 1 and user 2 decreases but it increases with user 3. A dynamic SIC receiver provides very low outage probability with the fixed SIC.	The outage probability increases with a back off factor of 10 dB.
Tang, Z. et al. [156]	Initially a slow fading channel was considered to define a outage achievable rate, then a statistical channel is used to estimate the SIC errors in a NOMA system.	Enhanced the outage achievable rate of NOMA system compared to the conventional OMA system and also achieved the higher data rates.	A perfect SIC receiver is assumed in this paper, but in practical cases an imperfect SIC needs to be considered to get accurate results.
Fan, J. et al. [157]	A single cell DL-NOMA system is considered for high speed railways in order to provide high data rates. A partial CSI with Rician fading channel is assumed to estimate the performance of NOMA system.	The proposed system enhanced the average outage probability compared to the conventional OMA system. The outage performance can be further increased by increasing the Rician factor *K*.	In high speed railways, Doppler shift arises due to the mobility, which leads to channel estimation errors and inter channel interference.
Liu, S. et al. [158]	Multi user DL-NOMA system with limited CSI feedback channel is considered to achieve high sum rate. Random beam-forming and zero forcing techniques are also considered to reduce the interference between the users.	Random beam-forming technique is more suitable for a limited CSI feedback channel and it achieves higher system sum rate than compared to the OMA system.	In this paper, only limited CSI feedback channel is considered, which reduces the overall system performance.
Saxena, P. et al. [159]	The effect of beam-forming technique in a multi input single output up-link NOMA system is considered with one bit feedback to estimate the system performance.	Enhanced the system performance and also achieved the high coding gain.	The overall performance of proposed system is reduced with the enhanced number of transmitting antennas.
Yang. Z. et al. [161]	A single cell DL-NOMA system performance is studied under imperfect CSI and CSI based on order statistics. Analytical expression for outage probability and average sum rate are derived.	Enhanced the performance of NOMA system compared to the conventional OMA system under two proposed CSI scenarios. Average sum rate and outage probability are enhanced with the SNR.	Perfect CSI gives better performance compared to the proposed two CSI scenarios because it is an ideal case.
Choi, J. [162]	A repetition based NOMA system is considered to achieve high diversity gain. An analytical expression for outage probability was derived based on the key parameter.	The proposed method enhanced the average error probability and outage probability.	The average error probability depends on the code length.

**Table 14 sensors-23-02356-t014:** Summary of optimization and analytical algorithms used to resolve the security issues in NOMA systems.

Author & Ref. No.	Approach	Advantages	Limitations
He, G. et al. [163]	An optimal PA and power splitting ratio selection algorithms are used to maximise the security sum rate (SSR) of the physical layer in a DL-NOMA system.	Enhanced the overall system performance with the optimal PA and power splitting ratio methods. The power splitting method provides the highest SSR compared to the PA method.	The SSR of the system is affected by the individual users’ minimum required energy.
Tang, J. et al. [164]	The security of the physical layer in NOMA system is affected by the random mobility of the users. In this paper, the authors considered random way point and direction mobility models to estimate the SSR.	Enhanced the SSR using the large protected zone radius. Users with good channel conditions will get higher SSR and fairness than compared with the users with poor channel conditions.	SSR is affected by the poor users. If the number of poor users increases, then the SSR and fairness decreases.
Yin, C. et al. [165]	UAV enabled multicast-unicast transmission system is considered for a DL-NOMA to maximise the SSR.	Enhanced the SSR of a UAV enabled NOMA system and also increased the uni-casting secrecy sum rate by increasing the transmission powers.	The multi-casting SSR decreased with the power because more powers are required to transmit the multi-cast information.
He, B. et al. [166]	Optimal NOMA scheme is designed to maximise the transmit power based on the SSR and QoS requirements.	Enhanced the performance of the NOMA system regardless of the number of users.	Eavesdroppers are assumed as passive in order to enhance the performance, but an eavesdropper will always deteriorate the overall performance.
ElHalawany, B.M. et al. [167]	A NOMA system with two users, i.e., trusted user and malicious user pairing is considered at the BS. Outage probability and secrecy outage probability are estimated for the proposed scheme.	The performance of the system depends on the users position from the BS. Enhanced secrecy outage probability is achieved when the malicious users are far away from the BS.	Secrecy outage probability deteriorates if the number of malicious users near the BS increases.
Liu, Y. et al. [168]	A technique called stochastic geometry is used to locate the users in NOMA system. Secrecy performance of the proposed technique is estimated based on single and multiple antennas.	Enhanced the secrecy performance by extending the eavesdroppers exclusion zone and using artificial noise at the BS.	Performance of the network is overestimated due to the assumption of perfect SIC.
Lv, L. et al. [169]	A novel transmission outage constrained scheme is considered in NOMA system to provide security and reliability. Outage probability and diversity order are estimated.	The proposed scheme provide the secured transaction without leakage of any confidential information	The increased number of eavesdroppers will reduce the efficiency.
Ziang, Z. et al. [170]	CR-NOMA network with multiple PUs and SUs is considered. Analytical expressions for outage probability and throughput are derived by assuming PU as a trusted user and SU as an eavesdropper.	Enhanced the secrecy performance by pairing the PUs with better channel conditions.	SUs will not improve the secrecy performance and throughput because they are assumed to be eavesdroppers.

**Table 15 sensors-23-02356-t015:** Summary of M-MIMO techniques used to enhance the data rates and capacity of 5G networks.

Author & Ref. No.	Approach	Advantages	Limitations
Suyama, S. et al. [176]	Massive MIMO technology was considered for the 5G and 6G networks to enhance the throughput and data rates.	High data rates of 100 Gbps are achieved using the mmWave frequency band of 28 GHz.	Multiple M-MIMO systems are used to enhance the throughput. This will increase the system complexity.
Tashiro, K. et al. [177]	A cylindrical antenna structure for the MIMO system was proposed to enhance the system capacity and SINR.	The system capacity is enhanced by 2.1 times compared to the planar structure. This will also enhance the coverage radius of up to 100 km.	Optimization of antennas will further enhance the system capacity.
Hussain, S.S. et al. [178]	Beam division multiple access technique was considered for the M-MIMO system to enhance the system capacity.	System capacity is enhanced 10 times and energy efficiency is enhanced 100 times by reducing the bit error rate.	Spectral efficiency and data rate enhancement is required to improve the overall system performance.
Zbairi, M. et al. [179]	A cell free MIMO UP receiver was considered with the zero forcing, MMSE, and maximum ratio combining detectors.	The spectral efficiency is enhanced using the zero forcing and MMSE detectors compared to the cellular M-MIMO and small cell M-MIMO systems.	The maximum ratio combining detector reduce the spectral efficiency.
Dicandia, F.A. et al. [180]	A triangular lattice structure of antennas was considered for M-MIMO system between the frequency bands of 24.25 GHz and 29.5 GHz.	Higher spectral efficiency is achieved by maintaining the minimum distance between the antennas.	The optimal distance should be estimated to improve the spectral efficiency and to achieve higher data rates.
Salh, A. et al. [181]	The trade-off between spectral efficiency and energy efficiency is used to reduce the transmit power based on multi-objective optimization in a M-MIMO system.	Spectral efficiency and energy efficiency are maximised by minimising total energy consumption.	The number of antennas is fixed in this paper.

**Table 16 sensors-23-02356-t016:** Summary of optimization algorithms used to enhance the coverage and capacity of 5G networks.

Author & Ref. No.	Approach	Advantages	Limitations
Campos, R.S. et al. [185]	A genetic algorithm is proposed to optimise the location identification of the mobile station in a cell and new BS deployment location.	Optimization of the mobile station location achieved lower latency and higher precision. Optimization of new BS deployment location enhanced the coverage and capacity of a cellular network.	Deploying new BSs in a lower cell density area leads to high positioning errors.
Thien, H.T. et al. [186]	Game theory based algorithm is used in D2D link pairs, to offload the data of one UE to another UE based on the offload coefficient. A Nash bargaining solution is considered to provide a game optimal solution.	Enhanced the energy efficiency and throughput by maximising the high D2D link fairness. The proposed method effectively offloads the UE data.	Energy efficiency decreases with the increase of minimum transmission power required, i.e., at 10 Mbps the energy efficiency is 26.56 and at 50 Mbps it is 24.21.
Cosma, I.S. et al. [187]	An innovative RL and neural network based scheduling algorithm is proposed to minimise the packet delay and drop rates by maintaining minimum QoS requirements.	The proposed algorithm enhanced the packet data rates and reduced the packet delays.	Performance of RL algorithm depends on the input parameters. Therefore, optimization network performance with respect to the input parameters is required.
Ahamad, M.M. et al. [188]	Higher order sectorization with proper antenna tilt is considered to enhance the coverage and capacity in 5G networks.	Phased antenna array enhanced the RF coverage at higher frequency bands compared to the low and mid frequency bands.	Accurate cell site planning is required to overcome the cell edge interference.
Sousa, M. et al. [189]	A new antenna has been proposed to optimise the antenna tilt and beam-forming antennas. The proposed model assumed that the antenna height at the BS should be less than the average height of the high rise buildings in order to enhance the coverage.	The proposed model enhanced the indoor and outdoor coverage area.	ML and RL algorithms are used to reduce the path loss and to enhance the accuracy.
Dandanov, N. et al. [190]	RL algorithm is considered to optimise the coverage and capacity based on the automatic antenna tilt angle.	The automatic antenna tilt will identify the higher traffic density locations and enhance the coverage and capacity by reducing the implementation cost and complexity.	Dynamic antenna tilt angle adjustment is required to avoid the coverage holes.
Qureshi, M.N. et al. [191]	Two RL based algorithms, deep Q-learning based on artificial neural networks and stochastic cellular learning, are used to optimise the antenna tilt.	Enhanced the overall coverage and data rates of a cellular network using two algorithms. ANN enhanced the cell edge performance compared to the stochastic method.	Handovers will increase the call rejection ratio, which will affect the overall system performance.
Dreifuerst, R.M. et al. [192]	Deep deterministic policy gradient and RL algorithms are used to enhance the coverage capacity and to minimise the interference by jointly optimising the antenna tilt and transmit power.	The proposed algorithms enhanced the coverage and capacity of a cellular network.	Automated optimisation techniques are required to predict the actual traffic density and coverage gaps.

**Table 17 sensors-23-02356-t017:** Summary of research articles based on performance indicators.

S.No	Performance Indicator	List of Research Articles
1	Coverage Capacity [177] $$C=\sum_{u=1}^{N_{u}}Blog_{2}\left(1+\frac{|h_{u}b_{u}{|}^{2}P_{u}}{P_{n}+\sum_{i=1,i\neq 4}^{N_{u}}{\left|h_{u}b_{i}\right|}^{2}P_{i}}\right)$$	[50,52,55,56,65,66,68,81,86,103,105,107,120,146,177,178,185,189,190,191,192]
2	Latency [90] $$E\left[d\right]=E\left[T_{q}\right]+E\left[c\right]+E\left[ans\right]$$	[89,90,91,92,93,94,95,96,185]
3	Throughput [57] $$TP_{u,r,k}=BR_{r}.(1-BLER\left(r,\gamma_{m(p),u,k}\right)$$	[52,56,57,67,69,81,82,84,85,89,96,99,101,102,103,104,106,120,122,139,140,141,186,187]
4	Spectral Efficiency [181] $$SE=log_{2}\left[\frac{1}{{\left(1+H^{H}H\right)}^{-1}}\right]$$	[54,56,67,84,86,99,101,108,120,150,179,181]
5	Data Rate [140] $$R_{u}\left(t+1\right)=\left(1-\frac{1}{\tau }\right)+\frac{r_{u}\left(t\right)}{T},u\in U$$	[69,99,100,121,134,140,143,146,176]
6	Outage Probability [168] $$P\left(R\leq R_{th}|B\right)=P\left(\sum_{i=1}^{B}R_{i}\leq R_{th}\right)$$	[50,54,68,83,155,156,157,158,161,162,167]
7	Interference Management [107] $$I=\sum_{j=1}^{N_{D}}P_{j}h_{j,e}d_{j,e}^{-\alpha }+\sum_{k=1}^{N_{s}}P_{k}h_{k,e}d_{k,e}^{-\alpha }+N_{0}$$	[57,78,79,87,98,106,107,110]
8	Power Consumption [150] $$P^{total}\left(s,\gamma \right)=\sum_{i=1}^{N_{F}}P_{i}^{total}\left(s,\gamma \right)$$	[50,82,150,151,152,153,178]
9	Overall System Performance [105] $$SINR_{D2DR}=\frac{|h_{TR}{|}^{2}P_{D}d_{TR}^{-\alpha }}{|h_{CR}{|}^{2}P_{C}d_{CR}^{-\alpha }+N_{0}}$$	[55,56,57,69,80,104,105,138,140,163,164,165,166,168,170,170]

Note: Only one mathematical expression is included for the 5G key parameters present in Table 17. However,
different authors follow different techniques to enhance these parameters, such as path loss techniques, SINR,
optimization algorithms, etc.

**Table 18 sensors-23-02356-t018:** Summary of 5G coverage enhancement techniques-advantages, limitations, and future challenges.

5G Techniques	Key Technologies	Benefits	Limitations	Challenges	References
Small cells	Femto Cells, Pico Cells, and Micro Cells	Enhanced capacity, enhanced throughput, enhanced coverage, and easy deployment	Reduced no. of small cells, infrastructure, expenditure, and coverage area	Coverage radius, mobility and handovers, deployment and testing	[49,50,52,53,54,56,57]
Carrier aggregation	Intra-band contiguous, intra-band non-contiguous, inter-band non-contiguous	Increased capacity, high data rates, improved load balancing, extended coverage, low latency	Battery life, coverage priority, proper filtering	Optimization algorithms to maximise the battery life, new CA techniques for the dynamic frequency bands, power amplifiers, filter design	[60,62,64,65,66,68,69]
D2D	Overlay mode, underlay mode, mode selection, device discovery	Enhanced capacity, spectrum sharing, interference management, inter-operability between DUE and CUE, cellular offloading	Security and privacy, limited range, malicious users	Initial device discovery, synchronisation, mode selection overhead, interference mitigation	[82,83,86,87,90,91,92,93,94,95,96,100,102,106,107,108,109,110]
NOMA	User pairing, power allocation, SIC decoder	High spectrum efficiency, low latency, massive connectivity, enhanced throughput	Receiver complexity, power efficiency	User pairing algorithms, SIC receiver algorithms, mobility	[120,121,122,123,131,132,133,134,135,136,137,140,142,143,151,152,153]
MIMO	MIMO, massive MIMO, multi user MIMO	Energy efficiency, throughput, spectral efficiency, higher data rates, channel capacity	System complexity, number of antennas, interference, power consumption	Signal detection, channel estimation, pilot contamination, energy efficiency	[176,177,178,179,180,181]
5G optimization	AI algorithms, ML and RL based algorithms	Enhanced coverage, low latency, high data rates	QoS, complexity	Bridging the gap between AI and 5G technologies with the use of AI and ML algorithms.	[184,185,186,187,188,189,190,191,192]

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
