# Peer review of "A Survey on 5G Coverage Improvement Techniques: Issues and Future Challenges"

_sensors, 2023, doi:10.3390/s23042356_

Round 1
Reviewer 1 Report
The survey examined the evaluation of mobile networks from 1G to 5G, its requirements, and the key parameters of 5G, including definitions of coverage and coverage requirements. It examined 5G coverage enhancement techniques, such as small cells, CA, D2D, NOMA, and optimization. This survey identifies and analyzes many recently published works. However, the following comments need to be addressed before recommending this paper for publication. The comments on this paper are as follows:
-
In section 1.1, the authors discuss existing surveys on 5G. It's good. But it is not clear how this paper is different from them. Would the authors please summarize the differences, as well as what additional information this paper provides? For example, refer to table 1 in 'Survey on recent advances in IoT application layer protocols and machine learning scope for research directions'.
-
The reader examines a survey paper to determine the key challenges left to tackle. This paper is missing that. It is recommended that the authors provide more details about open challenges in this field. There are several papers that discuss 6G, an advanced version of 5G, but the authors of this paper omit it from their paper.
-
AI and Machine Learning are emerging in this field. Therefore, authors should consider papers describing these areas, as well as optimization techniques. A new generation of smart communication models is emerging, so this discussion may be relevant.
-
The authors can provide a taxonomy for this paper to follow easily.
-
This paper lacks a graphic presentation. The authors must provide clear illustrations for better understanding.
-
For the published works, a statistical analysis is conducted. the emerging of this scope in the past and current situations are analyzed and plotted.
Author Response
Dear Professor,
Thank you very much for reviewing our manuscript and providing complimentary comments and suggestions. We have modified our article based on the suggestions and revised the manuscript accordingly.
Please find attached response and we hope that you find our responses satisfactory and that the manuscript is now acceptable for publication.

Reviewer 2 Report
The presented submission is a timely survey devoted to a very hot topic: coverage improvement techniques in 5G wireless networks.
The overall impression from the presented submission is quite positive. The text is well-written, well-structured, and well-formatted. The authors' logic and reasoning are clear.
Although, there are some issues, the resolving of which would improve the submission.
1. In the introduction, the authors mention the key technologies used in 5G. And at least one of them (MIMO/MU-MIMO/Massive-MU-MIMO) plays an important role in possible coverage management. But the authors skip this option and did not include it into their survey. It is strongly suggested to broaden the review by incorporating a subsection devoted to MIMO system coverage in 5G.
2. In my opinion, the survey lacks some generalization, that can shed light on the possible tracks for further research. Section 4 gives some hints, but very implicitly. The submission will benefit, if the authors unify the existing problems (maybe combining some of them), and show the possible technologies/approaches that can be assumed as the “growth points” for further development.
3. For a survey type of paper, the submission contains too many grammar and punctuation errors, e.g.:
— in line 135, “… with in…” → “… within…”;
— in line 147, “… effected by…” → “… affected by…”;
— in line 165, “… with respective to…” → “… with respect to …” or “…respective to …”;
— in line 171, “… un-successful …” → “… unsuccessful …”;
— in line 174, “… tools plays…” → “… tools play… ”;
— in line 209, “…cells uses…” → “… cells use… ”;
— in line 210, “… its low complexity…” → “…their low complexity…”;
— in line 222, “… colloids …” → “…collide … ”;
— in line 236, “… effective …” → “…effectiveness… ”;
— in line 257, “… which need …” → “…which needs… ”;
— in line 263, “… is very less compared …” → “…is less compared …”;
— in line 297, “… mode until unless the …” → “…mode until the…” or “…mode unless the… ;
— in line 323, “The wireless devices increases …” → “The number of wireless devices increases…”;
etc…..
So, please, revise the writing.
Author Response

(The authors gave the same response as above.)

Round 2
Reviewer 1 Report
1. Figure 2 is not informative. As of my knowledge, figure with only base station is meaningless.
2. Future Challenges are very limited in the paper. The authors are request to provide the future scope of the research.
3. The statistical analysis of the publications and future possibility to enhance this research are provided in the paper. For example: ref section 4 of 'Machine learning algorithms for wireless sensor networks: A survey' [not necessary to cite unless related]
4. There is a huge research going on "6G", but nothing found related to this regard. The authors may look in to it.
5. From Table 17, Mathematical formulae related to the performance measure used by the existing works are provided.
Author Response
Thank you very much for reviewing our manuscript. We also thank for providing complimentary comments and suggestions. We have revised the manuscript accordingly.

Reviewer 2 Report
The authors addressed most of my concerns and greatly improved the paper. It can be accepted in present form.
Author Response
Thank you very much for accepting the manuscript.